# Both LTA and LTB Subunits Are Equally Important to Heat-Labile Enterotoxin (LT)-Enhanced Bacterial Adherence

**DOI:** 10.3390/ijms24021245

**Published:** 2023-01-08

**Authors:** Qiangde Duan, Shengmei Pang, Lili Feng, Baoliang Li, Linfen Lv, Yuxuan Liang, Guoqiang Zhu

**Affiliations:** 1Department of Veterinary Microbiology, College of Veterinary Medicine, Yangzhou University, Yangzhou 225009, China; 2Jiangsu Co-Innovation Center for Prevention and Control of Important Animal Infectious Diseases and Zoonoses/Joint Laboratory for International Cooperation in Agriculture and Agricultural Product Safety, Ministry of Education, Yangzhou 225009, China; 3Jiangsu Joint Laboratory for International Cooperation in Prevention and Control of Important Animal Infectious Diseases and Zoonoses, Yangzhou 225009, China; 4Institute of Agricultural Economics and Information, Henan Academy of Agricultural Sciences, Zhengzhou 450002, China

**Keywords:** heat-labile enterotoxin, LTA subunit, LTB subunit, enterotoxigenic *Escherichia coli*, bacterial adherence

## Abstract

There is increasing evidence indicating that the production of heat-labile enterotoxin (LT) enhances bacterial adherence within in vitro and in vivo models. However, which subunit plays the main role, and the precise regulatory mechanisms remain unclear. To further elucidate the contribution of the A subunit of LT (LTA) and the B subunit of LT (LTB) in LT-enhanced bacterial adherence, we generated several LT mutants where their ADP-ribosylation activity or GM1 binding ability was impaired and evaluated their abilities to enhance the two LT-deficient *E. coli* strains (1836-2 and EcNc) adherence. Our results showed that the two LT-deficient strains, expressing either the native LT or LT derivatives, had a significantly greater number of adhesions to host cells than the parent strains. The adherence abilities of strains expressing the LT mutants were significantly reduced compared with the strains expressing the native LT. Moreover, *E. coli* 1836-2 and EcNc strains when exogenously supplied with cyclic AMP (cAMP) highly up-regulated the adhesion molecules expression and improved their adherence abilities. Ganglioside GM1, the receptor for LTB subunit, is enriched in lipid rafts. The results showed that deletion of cholesterol from cells also significantly decreased the ability of LT to enhance bacterial adherence. Overall, our data indicated that both subunits are equally responsible for LT-enhanced bacterial adherence, the LTA subunit contributes to this process mainly by increasing bacterial adhesion molecules expression, while LTB subunit mainly by mediating the initial interaction with the GM1 receptors of host cells.

## 1. Introduction

Enterotoxigenic *Escherichia coli* (ETEC) caused diarrhea continues to be an important cause of death in children, especially under 5 years old living in developing countries [1,2]. In addition, it is also a leading cause of bacterial diarrhea and mortality in young farm animals [3,4]. Two crucial types of virulence determinants produced by ETEC are adhesion molecules and enterotoxins, such as heat-labile enterotoxin (LT) and heat-stable enterotoxin (ST). To successfully establish infection and deliver the enterotoxins into mucosal intestinal epithelial cells, ETEC pathogens must first attach and adhere to these cells. Adhesion molecules are responsible for mediating the initial attachment and adhesion of ETEC to host cells, while enterotoxins promote cyclic AMP (cAMP) or cyclic GMP (cGMP) production, and ultimately induce fluid secretion, resulting in diarrhea. Given that ETEC is a kind of non-invasive pathogen, initial adhesion of the bacteria to host mucosal epithelial cells of the small intestine is necessary for ETEC infection caused diarrhea [5,6]. The importance of initial adherence in ETEC pathogenesis is well established, which virulence factors are involved and how they contribute to this crucial process remains unclear.

As one of the typical A_1_B_5_ enterotoxins, LT consists of one catalytic A subunit (LTA) and five B subunits (LTB) [7,8]. The catalytically active LTA subunit is comprised of A1 and A2 polypeptides linked by one disulfide bond. The A1 polypeptide with ADP-ribosylation activity is responsible for its enterotoxicity effects. The A2 polypeptide embeds the A1 polypeptide into the center of the LTB pentameric oligomer. The LTA subunit is responsible for stabilizing the LT holotoxin structure and accelerating the LTB subunit pentamer formation [9]. The LTB pentamer is required for binding the ganglioside GM1 receptors, lipopolysaccharide (LPS), and A-type blood sugar on the surface of host cells, and subsequently, entry into the intestinal epithelium cells [10,11,12]. The ability of LT to bind to different types of receptors on host cells (GM1 and A-type blood sugar) and LPS on the surface of *E. coli* depends on the specific amino acid residues in the LTB subunit [9]. A previous study reported that the secretion of most LT is still associated with outer membrane vesicles (OMVs) released by ETEC through the binding between LTB subunit and Kdo core of the LPS, and pre-treatment with soluble LPS was able to highly prevent LT binding to the bacterial surface of non-LT producing ETEC strains [11]. The Gly-33 residue is required for binding to GM1 receptors, while the Gln-3 and Thr-47 residues are required for binding to LPS [13,14,15,16]. Although the LTA and LTB subunits possess different molecular structures and biological properties, the stability of the LT holotoxin structure and the exertion of biological functions require synergistic effect of these two subunits.

Data from both in vivo and in vitro models has demonstrated that LT production plays a crucial role in enhancing bacterial initial adhesion to host cells beyond inducing diarrhea [17,18,19,20]. Secretion of LT favors the ETEC strains initial adherence and subsequent colonization of the intestinal epithelial cells in the mice and piglet models [17,18]. Furthermore, LT was also found to promote the adherence of ETEC and *Salmonella enterica* pathogens to intestinal epithelial cell lines in vitro [19,20]. Wang and his colleagues found that inhibiting the cAMP-dependent signal pathway blocked LT-increased bacterial adherence, suggesting LT contributing to the improvement in adherence ability of ETEC is mainly dependent upon its LTA subunit [21]. In contrast, another study found that the F4+ ETEC strain expressing the LTB subunit or the non-ADP-ribosylation activity mutant had a similar ability adherence to IPEC-J2 cells as the strain expressing LT holotoxin. Moreover, pre-treatment of IPEC-J2 cells cycloheximide had no influence on LT-enhanced bacterial adherence. The results suggested that the LTB subunit plays a predominant role in LT-promoted bacterial adherence [22]. It is clear that both subunits are important for LT-enhanced bacterial adherence, however, which subunit plays the central role and the underlying mechanism by which subunit contributes to this process remains controversial and poorly understood.

Some specific amino acid residues are necessary for the efficient function of LT. It has been reported that some monomeric mutant of the LTA subunit (S63K, A72R, R192G, and L211A) highly impairs its enterotoxicity [23,24,25,26]. In addition, another double mutant LTA subunit (R192G/L211A) was shown to just retain less than 0.1% of the enzymatic activity of LT [25]. Previous studies suggested that the ADP-ribosylation activity of the LTA subunit is necessary for LT-enhanced bacterial adherence [15,17]. To elucidate the precise mechanism of LTA and LTB subunits in promoting bacterial adherence, several LT mutants that impaired their ADP-ribosylation activity or receptor binding ability were constructed in this study. Furthermore, the abilities of the native LT and LT derivatives to enhance bacterial adherence were evaluated in vitro cell models. Our results indicated that both the ADP-ribosylation activity of LTA and GM1 receptor binding ability of LTB subunits contribute equally to LT-mediated enhancement of bacterial adherence, which provides insight into the pathogenic mechanism of LT and ETEC.

## 2. Results

### 2.1. Characterization of LT and LT Derivatives

To investigate which subunit predominantly contributes to LT-enhanced bacterial adherence, we generated the LTA subunit mutants LTA_R192G_, LTA_L211A_, and LTA_R192G/L211A_ (dmLT), and LTB subunit mutants LTB_Q3K_, LTB_G33S_, and LTB_T47A_. In contrast to the native LT, the intracellular levels of cAMP elicited by all the LTA_R192G_, LTA_L211A_, and dmLT mutants following incubation with IPEC-J2 cells were significantly reduced (Figure 1A). The levels of cAMP were induced following incubation with IPEC-J2 cells, and LTA_R192G_, LTA_L211A_ and dmLT mutants were just approximately 0.28%, 0.31%, and 0.07% of the level of cAMP induced by the native LT, respectively. GM1-binding ability results showed that the LTB_G33S_ mutant almost abolished its ability to bind to the GM1 receptor, and the ability of LTB_T47A_ mutant to bind GM1 receptor also significantly decreased compared to the native LT (Figure 1B). The GM1 receptor binding ability reduced approximately 95% in the LTB_G33S_ mutant, while it reduced approximately 25% in the LTB_T47A_ mutant. However, the LTB_Q3K_ mutant had a similar ability to bind to the GM1 receptor as the native LT. Moreover, non-LT producing *E. coli* strains 1836-2 and EcNc expressing native LT or LT mutants had similar growth rates, indicating the production of LT does not influence bacterial growth (Figure 1C,D). In addition, the native LT and LT mutants had a similar protein expression level in both *E. coli* 1836-2 (Figure 1E,F) and EcNc strains, suggesting that the mutation of single or double amino acid residues of LTA or LTB subunits would not affect the expression of LT.

### 2.2. LT Promotes E. coli Strains Adhesion to Intestinal Epithelial Cells

To confirm the role of LT in improving *E. coli* strains adhesion to epithelial cells, the adherence ability of two ETEC parent strains and their *ΔeltAB* isogenic mutants were compared in the in vitro cell models. The results revealed that the C83902 *ΔeltAB* mutant exhibited a significant reduction in the ability to adhere to IPEC-J2 intestinal epithelial cells when compared with the wild-type C83902 strain (Figure 2A, *p* < 0.01). Similarly, the H10407 *ΔeltAB* mutant exhibited a significant reduction in the ability to adhere to Caco-2 cells when compared with the wild-type H10407 strain (Figure 2B, *p* < 0.01). Quantification of bacterial colony forming units (CFUs) showed that the C83902 *ΔeltAB* and H10407 *ΔeltAB* mutants had approximately 87% and 74% reduction in adherence compared to the parent strains, respectively. Moreover, the adherence levels of the complemented strains (C83902 *ΔeltAB*/pLT and H10407 *ΔeltAB*/pLT) containing a recombinant plasmid expressing the full *eltAB* gene are similar to that of the wild-type strains (C83902 and H10407). These results confirmed that the production of LT plays a crucial role in enhancing *E. coli* strains adhesion to intestinal epithelial cells in vitro.

### 2.3. The ADP-Ribosylation Activity of LTA Subunit Is Required for LT-Enhanced Bacterial Adherence

To evaluate the role of the LTA subunit in LT-enhanced bacterial adherence, we compared the adherence ability of *E. coli* 1836-2 and EcNc strains expressing the native LT or LTA mutants that impaired the ADP-ribosylation activity. The adherence results revealed that non-LT producing *E. coli* 1836-2 and EcNc strains expressed either the native LT or LTA mutant had greater amount of bacteria adhesion to epithelial cells compared to the parent strains (Figure 3A,B). The adherence level was increased approximately 7.1- and 10.4-fold in the 2015 (1836-2/pLT) and 2022 (EcN/hLT) strains compared to the parent strains, respectively. While the adherence level was just increased approximately 2.4-, 2.5-, and 2.2-fold in 2019 (1836-2/pLTA_R192G_), 2020 (1836-2/pLTA_L211A_), and 2021 (1836-2/pLTA_R192G/L211A_) strains, respectively, and 3.8-, 3.9-, and 3.5-fold in the 2026 (EcN/hLTA_R192G_), 2027 (EcN/hLTA_L211A_), and 2028 (EcN/hLTA_R192G/L211A_) strains, respectively. The results suggested that the ADP-ribosylation activity of the LTA subunit plays a crucial role in LT-enhanced bacterial adherence.

### 2.4. Exogenously Supplied cAMP Improves Bacterial Adherence

Given the LTA mutants that impaired ADP enzymatic activity had a significant reduction in the ability to enhance bacterial adherence, we want to know whether the exogenous supply of cAMP can also improve bacterial adhesion to host cells. The results showed that exogenously supplied cAMP leads to a significant increase in *E. coli* 1836-2 and EcNc strains adhesion to IPEC-J2 or Caco-2 cells. The adherence ability of *E. coli* 1836-2 was increased approximately 3.2-, 6.7-, and 12.8-fold after being incubated with 2, 5, and 10 μM cAMP, respectively (Figure 4A). Likewise, the adherence ability of *E. coli* EcNc was increased approximately 4.1-, 8.6-, and 15.4-fold after treated with 2, 5, and 10 μM cAMP, respectively (Figure 4B). Moreover, the efficacy of exogenous cAMP on enhancing *E. coli* 1836-2 and EcNc strains adherence was eliminated after being pre-incubated with inhibitor Rp-cAMP (Figure 4C,D). The results indicated that the LTA subunit that contributes to LT-enhanced bacterial adherence was dependent on its ability to elevate the level of cAMP.

### 2.5. cAMP Increases the Expression of Bacterial Adhesion Molecules

Bacteria express adhesive molecules to promote the initial interaction with host cell receptors, which is beneficial for subsequent adhesion and colonization [27,28]. Thus, we speculated that the role of LTA in enhancing bacterial adherence may be mainly due to the increased expression of bacterial adhesion molecules. To determine whether cAMP elevated by LT would affect the adhesion molecules expressed by these two strains, *faeG* (encoding the major fimbrial subunit of F4 fimbriae), *fliC_H4_* (encoding the flagellin H4 serotype), *fimA* (encoding the major fimbrial subunit of type I fimbriae) of *E. coli* 1836-2, *fimA* (encoding the major fimbrial subunit of type I fimbriae), *focA* (encoding the major fimbrial subunit of F1C fimbriae), and *fliC_H1_* (encoding the flagellin H1 serotype) of *E. coli* EcNc were examined by qRT-PCR under stimulation with different concentrations of cAMP. The data revealed that the adhesion molecules expressed by either *E. coli* 1836-2 or EcNc highly increased after being treated with exogenous cAMP. The mRNA level of *faeG*, *fliC_H4_*, and *fimA* of *E. coli* 1836-2 was up-regulated approximately 3.6-, 2.7-, and 2.1-fold, respectively, in cells incubated with 5 μM cAMP, and approximately 6.3-, 4.7-, and 4.1-fold, respectively, in cells incubated with 10 μM cAMP (Figure 5A). Similarly, the mRNA level of *fimA*, *focA*, and *fliC_H1_* of *E. coli* EcNc was up-regulated approximately 4.7-, 1.9-, and 2.5-fold, respectively, in cells treated with 5 μM cAMP, and approximately 8.5-, 3.1-, and 4.1-fold, respectively, in cells treated with 10 μM cAMP (Figure 5B). These results suggested that cAMP is important to bacterial adherence.

### 2.6. Both GM1 and LPS Binding Abilities Are Required for LTB-Enhanced Bacterial Adherence

Previous studies have suggested that the Gly-33 residue of the LTB subunit is required for binding to GM1 receptors, while the Gln-3 and Thr-47 residues are required for binding to LPS on the surface of bacteria [19,20,21,22]. To explore the importance of the GM1 and LPS binding abilities of the LTB subunit for bacterial adherence, we constructed the LTB_Q3K_, LTB_G33S_, and LTB_T47A_ mutants, and compared their abilities to improve bacterial adherence with the native LT. The results showed that the adherence level was increased approximately 7.1-, 5.8-, 2.9-, and 3.8-fold in the 2015 (1836-2/pLT), 2016 (1836-2/pLTB_Q3K_), 2017 (1836-2/pLTB_G33S_), and 2018 (1836-2/pLTB_T47A_) strains compared to the non-LT producing parent 1836-2 strain, respectively (Figure 6A). Similarly, the adherence level of 2022(EcN/hLT), 2023 (EcN/hLTB_Q3K_), 2024 (EcN/hLTB_G33S_), and 2025 (EcN/hLTB_T47A_) strains was increased approximately 10.4-, 7.7-, 2.8-, and 5.6-fold compared to the non-LT producing parent EcNc strain, respectively (Figure 6B). These results indicated that both GM1 and LPS binding abilities of the LTB subunit are required for LT-enhanced bacterial adherence.

### 2.7. Membrane Cholesrerol Is Important for LTB–Mediated Bacterial Increased Adherence

Ganglioside GM1, the receptor for the LTB subunit, is enriched in lipid rafts. To examine whether lipid rafts are required for the LTB subunit-mediated enhancement of bacterial adherence, the membrane cholesterol was removed by Methyl-β-cyclodextrin (MβCD). The cytotoxicity results showed that IPEC-J2 cells treated with less than 10 mM MβCD has no apparent effect on cell viability (*p* > 0.05) (Figure 7A). Flow cytometry analysis results showed that approximately 30% of cholesterol was depleted in IPEC-J2 cells after being treated with 10 mM MβCD (Figure 7B). Moreover, treating IPEC-J2 cells with MβCD significantly reduced the adhesion of 2015 (1836-2/pLT), 2016 (1836-2/pLTB_Q3K_), and 2018 (1836-2/pLTB_T47A_) strains to cells by approximately 37%, 31%, and 20%, respectively (Figure 7C). Complementation with exogenous cholesterol restored bacterial adhesion to similar levels of untreated cells. However, treatment with MβCD did not significantly reduce the adhesion of wild-type 1836-2 and 2017 strains. These results suggested that the GM1 binding ability and membrane cholesterol play a crucial role in LTB-enhanced bacterial adherence.

## 3. Discussion

Despite it being well established that LT plays a role in enhancing enteric bacterial adherence, the precise molecular mechanisms of its two subunits remain to be determined. Previous studies investigated the role of the LTA or LTB subunit in promoting bacterial adherence mainly by deletion of one subunit at a time or by expressing one subunit alone in the non-LT producing strains [19,21,22]. The results can not exactly reflect the function of each subunit in a normal physiological state due to the inability to form the functional LT holotoxin. In this study, the single or double LT mutants were generated by site-directed mutagenesis of the key specific amino acid residues, which just impaired the efficient function of the LTA or LTB subunit but would not affect the structure of the LT holotoxin. Therefore, the obtained results in our study may be more reliable than the previous reports.

Consistent with previous findings, we found that deletion of *eltAB* gene from the porcine-derived ETEC C83902 strains and human-derived H10407 strains resulted in their adherence abilities being significantly decreased compared with the wild-type strains [13,14,15]. The results confirmed that production of LT is able to improve ETEC strains adhesion to host cells in vitro. To further elucidate the mechanism by which LT contributes to promoting bacterial adherence, we used two non-LT producing *E. coli* strains to express either native LT or LT mutants, and their adherence abilities were evaluated in IPEC-J2 or Caco-2 cell models. Compared to the native LT, the LT mutants had a significantly reduced ability to enhance the non-LT producing *E. coli* 1836-2 and EcNc strains adhesion to host cells, suggesting both the LTA and LTB subunits are required for LT-mediated enhancement of bacterial adherence. Besides the attenuated enterotoxicity LTA mutants and the impaired GM1 binding ability, LTB mutants showed similar abilities to improve bacterial adherence, indicating both subunits play an equally role in LT-enhanced bacterial adherence.

The functional holotoxin LT is comprised of a single LTA subunit and a pentameric LTB subunit. The specific binding between the LTB subunit and GM1 receptors on the host cell surface is required for entry into intestinal epithelial cells, and the ADP-ribosylation activity of the LTA is required for elevating cAMP secretion [10,29]. Disruption of the holotoxin will result in loss of its function. In this study, our data showed that the strains expressing the LTA mutants have no enhancement of cAMP concentration (Figure 1A), they still caused certain increases in the bacterial adherence. However, the abilities of the three LTA mutants to promote bacterial adherence were significantly lower than that of the native LT. The results indicated that not only the ADP-ribosylation activity of the LTA subunit, but also the LTB subunit, is required for LTA mediated enhancement of bacterial adherence. The results confirmed that strains carrying the plasmids expressing the LTB subunit mutants also still increased the bacteria adhesion to host cells (Figure 6). The LTA subunit-induced cAMP is an important cellular “second messenger” that regulates pathogen’s virulence factors in multiple ways. Previous results demonstrated that the elevated cAMP improved the expression of K88, 987P, and type 1 fimbriae [19,30,31]. In this study, we found that the expression of K88 fimbriae, type Ⅰ fimbriae, flagella of *E. coli* 1836-2, type Ⅰ fimbriae, F1C fimbriae, and flagella of *E. coli* EcNc was significant up-regulated after treatment with exogenous cAMP. Therefore, our results indicated that the LTA subunit promoting bacterial adherence is mainly dependent on its ability to elevate the cAMP levels, which leads to up-regulated expression of bacterial adhesion molecules.

The mammalian membrane microenvironment is important for pathogens attachment to host cells [32,33]. The interaction of the LTB subunit and its GM1 receptor is required for the LT’s typical internalization pathway leading to toxic effects. In this study, our data showed that the *E. coli* 1836-2 and EcNc strains expressed as the LTB subunit mutants (Q3K, G33S, and T47K) had a greater amount of adhesion to host cells than the parent strains. However, the ability of the attenuated GM1 binding mutant (G33S) to promote bacterial adhesion was significantly lower than that of native LT. Deletion of cholesterol from epithelial cells by MβCD, and the ability of native LT, LTB_Q3K_, and LTB_T47A_ mutants to promote bacterial adherence was significantly reduced. However, it did not impair the ability of LTB_G33S_ mutant to enhance *E. coli* 1836-2 adhesion to IPEC-J2 cells. Our results suggested that LTB-enhanced bacterial adherence is mainly dependent on its GM1 binding activity. Interestingly, the Q3K and T47K mutants with known impairment in binding to LPS also showed a lower ability to promote bacterial adherence than the native LT, indicating the LPS binding activity is also involved in LTB-enhanced bacterial adherence. Therefore, the LTB subunit contributes to increasing bacterial adherences, which predominantly depends on its GM1 receptor binding activity, but also on its LPS binding activity. The LTB subunit could regulate LTB-enhanced bacterial adherence in two different ways. First, the interaction of the LTB subunit with GM1 receptors on the surface of host cells facilitates the internalization of the LTA subunit in which the LTA exerts its enterotoxicity. Besides, the LTB subunit binding to the LPS in OMVs located on the bacterial surface would form an “OMVs-LTB-GM1” bridge between the bacteria and host cells to improve subsequent adherence [11,19].

In conclusion, our results demonstrate that the LTA and LTB subunits are required for LT-enhanced bacterial adherence, and both subunits play an equally important role in this process. Moreover, the LTA subunit contributes to increasing bacterial adherence and is dependent on its ability to elevate intracellular cAMP levels. Elevated cAMP will increase the expression of bacterial adhesion molecules that are crucial for the initial attachment and subsequent colonization. The released cAMP elicited by LT can regulate the expression of bacterial adhesion molecules, in turn suggesting a cross-talk in the expression of virulence determinants when pathogens cause infections. Both its ability to bind host cells’ GM1 receptors and LPS on the surface of bacteria are required for LTB-enhanced bacterial adherence. The binding with GM1 receptors would favor its LTA subunit internalized into host cells to elevate the cAMP production and activate the cAMP-dependent cellular events. Association with LPS on the surface of *E. coli* would make the LT function as a bridge to facilitate the attachment of the bacteria and host cells, leading to adherence increased. Our results will advance our understanding of the pathogenic mechanisms of LT and provide new strategies for developing new vaccines and drugs to prevent ETEC infection.

## 4. Materials and Methods

### 4.1. Bacterial Strains, Plasmids, and Cell Line

Bacterial strains and plasmids used in this study are included in Table 1. The ETEC C83902, a wild-type strain isolated from piglet diarrhea [34], and wild-type ETEC H10407, a strain isolated from human diarrhea [35], were used as the DNA template to PCR amplify the porcine *eltAB* gene and human *eltAB* gene, respectively. The 1836-2 strain is a nonpathogenic ETEC field isolate that is used for expressing porcine LT and its derivatives [14]. The EcNc strain is derived from Nissle 1917 deletion of its two cryptic plasmids, pMUT1 and pMUT2, that are used for expressing human LT and its derivatives [36]. The wild-type *E. coli* C83902, 1836-2, and EcNc were cultured in antibiotic-free Luria-Bertani (LB) broth or LB-agar plates [18,34,36]. The 1836-2 and EcNc strains harboring the native LT or LT derivatives plasmids were grown in LB with ampicillin (Amp) at the concentration of 100 μg/mL.

The porcine intestinal epithelial cell line IPEC-J2, derived from neonatal porcine jejunum, and the Caco-2 cell line, derived from human colorectal adenocarcinoma, were cultured in Dulbecco’s minimal Eagle medium (DMEM), supplemented with 10% (*v*/*v*) heat-inactivated fetal bovine serum (FBS). Cells were maintained in a humidified incubator with 5% (*v*/*v*) CO_2_ at 37 °C.

### 4.2. Construction of LT and LT Derivatives

The full-length *eltAB* gene was amplified by PCR from ETEC C83902 or H10407 strains whole genomic DNA using the primers pBR-LT-F and pBR-LT-R (Table 2). Splicing by an overlapping extension (SOE)-PCR was used to construct the LTB_Q3K_, LTB_G33S_, LTB_T47A_, LTA_R192G_, LTA_L211A_, and LTA_R192G/L211A_ double mutant (dmLT) site-directed mutants using the specific primers listed in Table 2. Using the LTB_Q3K_ derivative as an example, two DNA fragments were initially PCR amplified with primers pBR-LT-F/LTB_Q3K_-R and LTB_Q3K_-F/pBR-LT-R. Then, SOE-PCR was used to overlap the two fragments with primers pBR-LT-F/pBR-LT-R. The LTB_G33S_, LTB_T47A_, LTA_R192G_, LTA_L211A_, and dmLT derivatives were constructed using the same method. Each gene was digested with *Eco*RV and ligated into the pBR322 vector. The plasmids successfully constructed were confirmed using restriction digestion and DNA sequencing.

### 4.3. Quantitative Measurement of cAMP Concentration

The cAMP concentration was tested by competitive ELISA using a cAMP Parameter assay kit (R&D Systems, Minneapolis, MN, USA) according to the manufacturer’s instructions. Briefly, 2015, 2019, 2020, 2021, 2026, 2027, and 2028 strains were grown overnight and adjusted to OD_600_ = 1.0 using LB. Bacterial culture was diluted 1:100 to fresh LB broth (containing 100 μg/mL Amp) and grown for 16 h in a 37 °C shaker. Then, 100 μL of the filtered overnight culture medium from 2015, 2019, 2020, and 2021 strains was added to each well of IPEC-J2 cells in a 6-well tissue culture plate. Additionally, 100 μL of the filtered overnight culture medium from 2022, 2026, 2027, and 2028 strains was added to each well of Caco-2 cells in a 6-well tissue culture plate. The cells were rinsed three times in cold phosphate-buffered saline (PBS) after being cultured at 37 °C for 1 h. Cells were re-suspended in a 500 μL cell lysis buffer, and cellular debris was removed by centrifuge at 1000 rpm for 10 min at 4 °C. Cell lysates were examined for intracellular cAMP level by following the manufacturer’s protocol.

### 4.4. GM1-ELISA Assay

The GM1-binding activity of LT and LT derivatives was tested by GM1-ELISA as described previously [37]. Briefly, the 2015, 2016, 2017, 2018, 2022, 2023, 2024, and 2025 strains overnight bacterial culture was initially adjusted to the same optical density value (OD_600_ = 1.0), then the overnight cultured bacterial culture was diluted 1:100 to 5 mL of fresh LB solution containing 100 μg/mL Amp and grown for 16 h at 37 °C with vigorous shaking. Subsequently, 100 μL of the filtered culture medium from each strain was added to a 96-well plate coated with 400 ng/well of GM1 (Sigma, St. Louis, MO, USA) as the antigen and incubated at 37 °C for 1 h. Subsequently, 100 μL of anti-CT antibody (1:1000) as the primary antibody was incubated for 1 h and washed three times with PBS + 0.05% (*v*/*v*) Tween 20 (PBST). Then, 100 μL of HRP-conjugated goat anti-rabbit IgG (1:5000) as the secondary antibody was incubated for an additional 1 h at 37 °C. A 50 μL stop solution was added to each well after incubation with 3.3′5.5′ tetramethylbenzidine (TMB) substrate (Beyotime, Shanghai, China) at room temperature for 30 min. Finally, the OD of each well was determined using a microplate reader at 450 nm.

### 4.5. Bacterial Growth Curve Determination

*E. coli* 1836-2, EcNc, 1836-2/p*eltAB*, 1836-2/pLTB_Q3K_ 1836-2/pLTB_G33S_, 1836-2/pLTB_T47A_, 1836-2/pLTA_R192G_, 1836-2/pLTA_L211A_, 1836-2/pLTA_R192G/L211A_, EcNc/hLTB_Q3K_, EcNc/hLTB_G33S_, EcNc/hLTB_T47A_, EcNc/hLTA_R192G_, EcNc/hLTA_L211A_, and EcNc/hLTA_R192G/L211A_ strains were grown overnight in glass culture tubes at 37 °C on a shaker operated at 200 rpm. The next day, 40 μL of each overnight bacterial culture was sub-cultured in 4 mL of fresh LB medium, and the OD_600_ was measured hourly over an 8-h period.

### 4.6. Western Blotting Analysis

LT and LT derivatives expression in *E. coli* 1836-2 or EcNc were measured by immunoblotting, as described previously [38]. Briefly, strains from 2015, 2016, 2017, 2018, 2019, 2020, and 2021 were incubated in Casamino acids-Yeast extract (CAYE) medium for 16 h, the bacterial culture was then adjusted to an OD_600_ of 1.0. 1.5 mL and the bacterial culture was centrifuged at 12,000 rpm, 4 °C for 10 min. The cell pellet was collected and resuspended in 80 μL sterile PBS and 20 μL 5 × SDS loading buffer. The whole-cell samples were boiled for 10 min and separated by sodium-dodecyl-sulfate-polyacrylamide gel electrophoresis (SDS-PAGE), and then transferred onto a polyvinylidene difluoride membrane (PVDF). After being blocked by 5% skim milk in PBST, the membrane was incubated with the rabbit anti-CT antibody (1:8000, Sigma, St. Louis, MO, USA) as the primary antibody and HRP-labeled goat anti-rabbit immunoglobulin IgG (1:10,000, Abclonal, Wuhan, China) as the secondary antibody. The blots were developed using an enhanced chemiluminescent (ECL) kit (NCM-Biotech, Suzhou, China) with the ChemiDoc^TM^ Imaging system (Bio-Bad, Hercules, CA, USA). The ImageJ version 1.47v software (Wayne Rasband, National Institutes of Health, Bethesda, MD, USA) was used to quantify the images. The relative protein levels of LT and its mutants were calculated. The protein level of LT of the 2015 strain was assumed as 100%, and all LT mutants were normalized to LT of the 2015 strain. Data are expressed as the mean ± SD of three independent experiments.

### 4.7. Generation of eltAB Isogenic Mutants

The C83902*ΔeltAB* and H10407*ΔeltAB* isogenic mutants were generated using the λ-Red recombinant method, described previously [39]. Briefly, the chloramphenicol (Cm+) cassette was amplified from the plasmid pKD3 by PCR using the *ΔeltA*B-F and *ΔeltAB*-R primers (Table 2) that is homologous to sequences within 5′ and 3′ regions of the *eltAB* gene. The purified PCR product was transferred into the *E. coli* C83902 and H10407 strains containing plasmid pKD46. Positive primary recombinants were selected on LB plates containing both Cm+ and Amp+ antibiotics. Finally, plasmid pCP20 that encoded the FLP recombinase was introduced to delete the Cm+ cassette. Successful deletion of the *ΔeltAB* gene in the isogenic mutants was confirmed by DNA sequencing, GM1-, and cAMP-ELISA assays.

### 4.8. Bacterial Adherence Assay

IPEC-J2 and Caco-2 cells were seeded in 24-well plates and grown to a confluent monolayer. Bacteria were grown overnight in 5 mL of LB or LB medium supplemented with Amp+ (100 μg/mL) at 37 °C. Then, the overnight bacterial culture was diluted to 5 mL fresh medium and continued to grow for 3 h. Bacteria were then added to each well of IPEC-J2 or Caco-2 cells at a multiplicity of infection (MOI) ratio of 30:1. After 1.5 h incubation, cells were gently washed three times with sterile PBS to remove the non-adherent bacteria. Cells with adherent bacteria were dislodged by treating with 1 mL of 0.25% (m/v) trypsin for 20 min at 37 °C. The suspensions were mixed well, serially diluted (1:10), and plated on LB plates. Bacteria CFUs on overnight growth LB plates were counted.

For exogenously supplied cAMP effects on the adherence ability of *E. coli* 1836-2 and EcNc, each of these two strains were grown under LB medium containing 0, 2, 5, or 10 μM cAMP for 3 h to a logarithmic growth phase. Then, infected IPEC-J2 or Caco-2 cells were pre-incubated with or without 200 μM Rp-cAMP (an inhibitor of PKA). The number of bacteria adhering to cells was determined as described above.

### 4.9. RNA Extraction and Real-Time Quantitative PCR (RT-qPCR)

*E. coli* 1836-2 and EcNc were cultured in LB medium containing 0, 2, 5, or 10 μM cAMP for 3 h. Total RNA was extracted from the exponential culture bacterial samples using TRIzol reagent (TIANGEN, Beijing, China), as previously described [38]. The yield and quality of RNA were determined by agarose gel electrophoresis and a Nanodrop2000 spectrophotometer. The cDNA was synthesized using a FastKing gDNA Dispelling RT SuperMix Kit (TIANGEN, Beijing, China) according to the manufacturer’s instructions. The endogenous housekeeping genes glyceraldehyde-3-phosphate dehydrogenase (*GAPDH*) and *cysG* were chosen as the reference gene. Specific primers for amplification of *faeG*, *fimA*, *focA*, *fliC_H1_*, and *fliC_H4_* genes are listed in Table 2.

RT-qPCR amplification was performed using a AceQ qPCR SYBR Green Master Mix (Vazyme, Nanjing, China) according to the manufacturer’s instructions. For each sample, 100 ng of cDNA and 300 nM concentrations of each primer set were mixed with 10 μL of AceQ qPCR SYBR Green Master Mix per well. A two-step program was run in the 7500 Real-time system (Applied Biosystems, CA, USA). Both reference genes had a stable expression in all samples tested. The data from bacteria were normalized to the two endogenous reference genes *GAPDH* and *cysG*. The results were analyzed using the 2^−ΔΔCT^ method.

### 4.10. MβCD Cytotoxicity

The membrane cholesterol of IPEC-J2 cells was deleted by treating with different concentrations of MβCD (Sigma, St. Louis, MO, USA). The cell viability of IPEC-J2 cells after treating with MβCD was quantitated using a Cell Counting Kit-8 (Absin, Shanghai, China), as previously described [40]. Briefly, 100 μL of DMEM medium containing different concentrations of MβCD (0–20 mM) was added to each well of IPEC-J2 cells in the 96-well plate and incubated for 1 h at 37 °C. Then, 10 μL of CCK8 solution was added to each well. The absorbance values of the samples were read at 450 nm using a microplate reader. The percentage viability of cells was calculated with respect to that of the untreated control cells (set as 100%). Each concentration group had 12 replicate wells, and the experiment was performed three times.

### 4.11. Flow Cytometry Analysis of Efficiency of Cholesterol Removal

GM1 ganglioside has been widely used as a marker of membrane lipid rafts. The efficiency of cholesterol removal by MβCD was performed by detecting CTB-FITC conjugated (Absin, Shanghai, China) binding to GM1 using flow cytometry. IPEC-J2 cells were seeded into a 6-well tissue culture plate and grown to 100% confluence. Cells were treated with 10 mM MβCD for 1 h at 37 °C, and then digested with 0.25% (m/v) trypsin at 37 °C for 20 min. The detached cells were further incubated with CTB-FITC antibodies at the concentration of 10 μg/mL for 30 min on ice. Flow cytometry was carried out using a CytoFlex S flow cytometer (BECKMAN COULTER, CA, USA) with 10,000 cells collected, and data were analyzed using FlowJo soft-ware, version10.7.2 (BD Biosciences, Ashland, OR, USA). Each sample had 3 replicate wells, and the experiment was performed three times.

### 4.12. Effect of Lipid Rafts on LTB-Enhanced Bacterial Adherence

Each well of IPEC-J2 or Caco-2 cells in the 24-well plate was pre-treated with 100 μL of 10 mM MβCD for 1 h at 37 °C and washed with PBS three times. Then, cells were infected at a MOI of 30 with bacteria either supplemented with or without 400 μg/mL cholesterol (St. Louis, MO, USA) for an additional 1.5 h. The number of bacteria adhering to cells was determined as described above.

### 4.13. Statistical Analysis

Student’s t test and one-way ANOVA statistical analysis were used for statistical comparisons of differences between two or more than two groups, respectively. Data are expressed as the mean with the SD. * indicates *p* < 0.05 and is deemed statistically significant. ** indicates *p* < 0.01 and is deemed statistically highly significant.

## Figures and Tables

**Figure 1 ijms-24-01245-f001:**
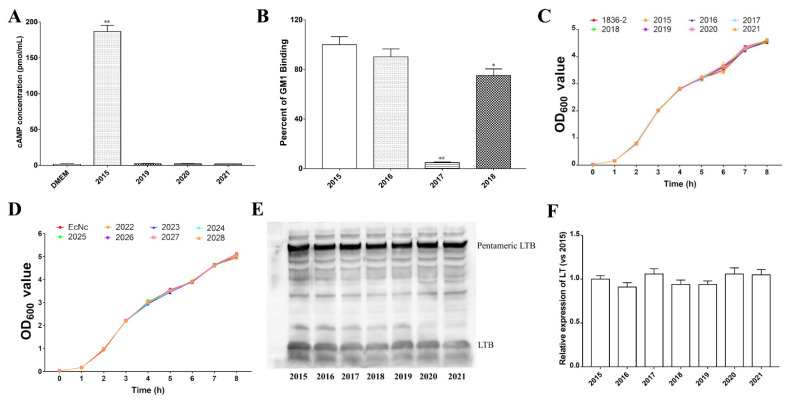
Characterization of native LT and its mutants. (**A**) The intracellular levels of cAMP induced by 2015 (1836-2/pLT), 2019 (1836-2/pLTA_R192G_), 2020 (1836-2/pLTA_L211A_), and 2021 (1836-2/pLTA_R192G/L211A_) strains following incubation with IPEC-J2 cells. The intracellular level of cAMP from IPEC-J2 cells using just incubation with DMEM medium used as the negative control. (**B**) The GM1-binding activity of the culture medium from 2015, 2016 (1836-2/pLTB_Q3K_), 2017 (1836-2/pLTB_G33S_), and 2018 (1836-2/pLTB_T47A_) strains. The GM1-binding activity of the culture medium from 2015 strain was presented as 100%. Data are expressed as the mean ± standard deviation (SD) of triplicate independent experiments. (**C**) Growth curves for *E. coli* strains 1836-2, 2015, 2016, 2017, 2018, 2019, 2020, and 2021. (**D**) Growth curves for *E. coli* strains EcNc, 2022 (EcN/hLT), 2023 (EcN/hLTB_Q3K_), 2024 (EcN/hLTB_G33S_), 2025 (EcN/hLTB_T47A_), 2026 (EcN/hLTA_R192G_), 2027 (EcN/hLTA_L211A_), and 2028 (EcN/hLTA_R192G/L211A_). Data are expressed as the mean ± standard deviation of quadruplicate independent experiments. (**E**) Immunoblotting of LT and its mutant’s expression in *E. coli* 1836-2 strain. (**F**) Quantification of relative LT and LT mutants’ levels shown in panel E. * indicates *p* < 0.05 and is deemed statistically significant. ** indicates *p* < 0.01 and is deemed statistically highly significant.

**Figure 2 ijms-24-01245-f002:**
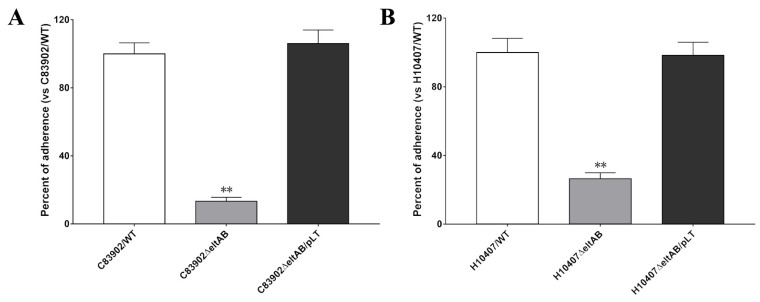
Adherence percentage of differing ETEC strains to intestinal epithelial cells. (**A**) Adhesion of ETEC C83902/WT, C83902 *ΔeltAB*, and C83902 Δ*eltAB*/pLT strains to IPEC-J2 cells. The adhesion index of the C83902/WT strain was assumed to be 100%. (**B**) Adhesion of ETEC H10407/WT, H10407*ΔeltAB*, and H10407 Δ*eltAB*/pLT strains to Caco-2 cells. The adhesion index of the H10407/WT strain was assumed to be 100%. Data are expressed as mean ± SD of triplicate experiments. ** indicates *p* < 0.01 and is deemed statistically highly significant.

**Figure 3 ijms-24-01245-f003:**
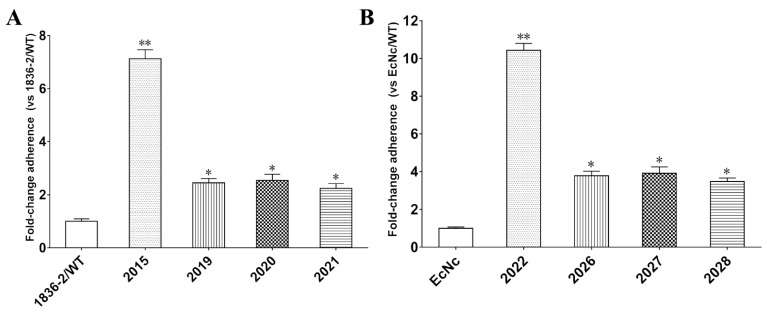
The ability of native LT or LTA mutants mediated *E. coli* 1836-2 and EcNc adhesion to intestinal epithelial cells. (**A**) Adhesion of ETEC 1836-2 expressed native LT (strain 2015) or LTA mutants (strains 2019, 2020, and 2021) to IPEC-J2 cells. The adhesion index of wild-type ETEC 1836-2 was assumed to be 100%. (**B**) Adhesion of *E. coli* EcNc expressed native LT (strain 2022) or LTA mutants (strains 2026, 2027, and 2028) to Caco-2 cells. The adhesion index of parent *E. coli* EcNc was assumed to be 100%. Data are expressed as mean ± SD of triplicate experiments. * indicates *p* < 0.05 and is deemed statistically significant. ** indicates *p* < 0.01 and is deemed statistically highly significant.

**Figure 4 ijms-24-01245-f004:**
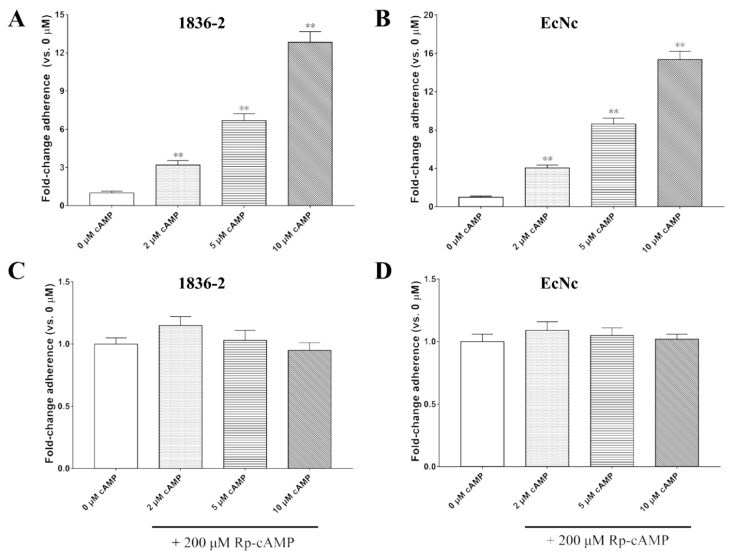
Exogenously supplied cAMP enhances bacterial adherence in a concentration-dependent manner. (**A**) Pre-incubated with 2, 5, and 10 μM cAMP increases the adherence ability of 1836-2 strain adhesion to IPEC-J2 cells in a concentration-dependent manner. (**B**) Exogenously supplied with 2, 5, and 10 μM cAMP promotes *E. coli* EcNc strain adhesion to Caco-2 cells. (**C**) Pre-treated 200 μM Rp-cAMP block the efficacy of cAMP on promoting ETEC 1836-2 strain adhesion to IPEC-J2 cells. (**D**) Pre-treated with 200 μM Rp-cAMP block efficacy of cAMP on promoting *E. coli* EcNc strain adhesion to Caco-2 cells. Data are expressed as mean ± SD of triplicate experiments. ** indicates *p* < 0.01 and is deemed statistically highly significant.

**Figure 5 ijms-24-01245-f005:**
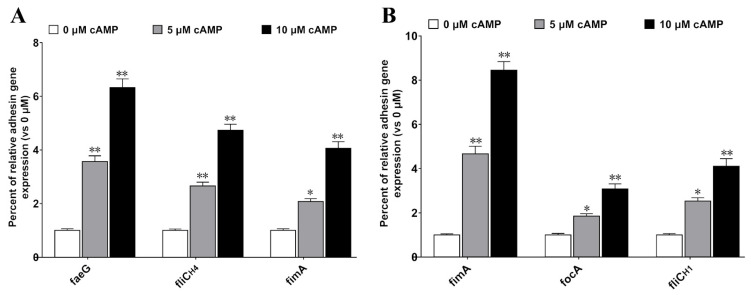
Exogenously supplied 5 μM and 10 μM cAMP up-regulates the adhesion molecules mRNA expression of *E. coli* 1836-2 and EcNc. (**A**) Treatment with cAMP increases the expression of *faeG*, *fliC_H4_*, and *fimA* genes of ETEC 1836-2. The mRNA level of *faeG*, *fliC_H4_*, and *fimA* without cAMP treatment was considered 100%. (**B**) Treatment with cAMP increases the expression of *fimA*, *focA*, and *fliC_H1_* genes of *E. coli* EcNc. The mRNA level of *fimA*, *focA*, and *fliC_H1_* without cAMP treatment was considered 100%. The data are expressed as the mean ± SD of four independent experiments. * indicates *p* < 0.05 and is deemed statistically significant. ** indicates *p* < 0.01 and is deemed statistically highly significant.

**Figure 6 ijms-24-01245-f006:**
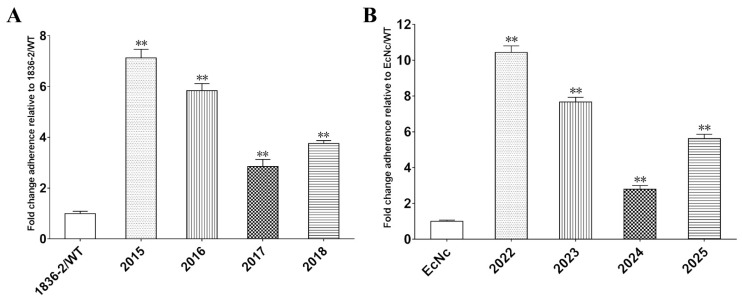
Adhesion of *E. coli* 1836-2 or EcNc expressed native LT or LTB subunit mutants to IPEC-J2 cells or Caco-2 cells. (**A**) Adhesion of ETEC 1836-2 strain expressed native LT (strain 2015) or LTB mutants (strains 2016, 2017, and 2018) to IPEC-J2 cells. The adhesion index of ETEC 1836-2 strain was assumed to be 100%. (**B**) Adhesion of *E. coli* EcNc strain expressed native LT (strain 2022) or LTB mutants (strains 2023, 2024, and 2025) to Caco-2 cells. The adhesion index of *E. coli* EcNc strain was assumed to be 100%. Data are expressed as mean ± SD of triplicate independent experiments. ** indicates *p* < 0.01 and is deemed statistically highly significant.

**Figure 7 ijms-24-01245-f007:**
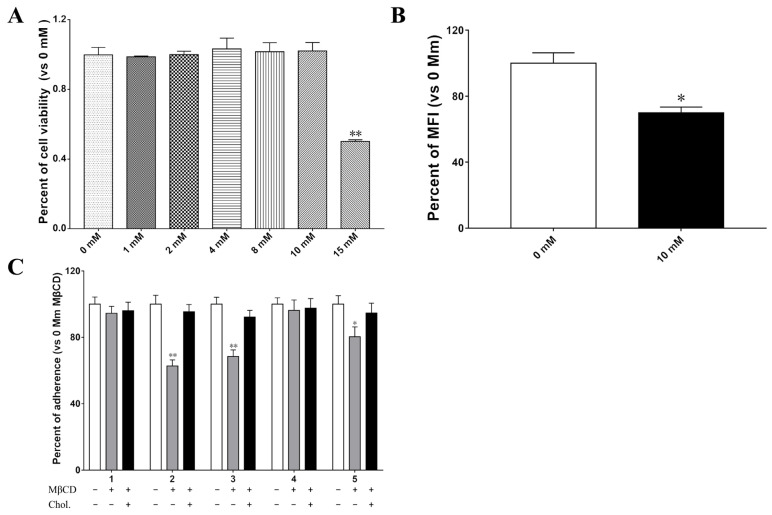
Membrane cholesterol is important for LT-increased ETEC 1836-2 adhesion to IPEC-J2 cells. (**A**) Different concentrations of MβCD effects on cell viability of IPEC-J2 cells. The viability of untreated IPEC-J2 was assumed to be 100%. (**B**) Cholesterol removal efficiency after treating IPEC-J2 cells with 10 mM MβCD. The mean fluorescence intensity (MFI) of the untreated IPEC-J2 cells was assumed to be 100%. (**C**) Pre-treated IPEC-J2 cells with MβCD blocks LTB-enhanced bacterial adherence. The adhesion index of ETEC 1836-2 strain to untreated IPEC-J2 cells was assumed to be 100%. Data are expressed as mean ± SD of triplicate independent experiments. * indicates *p* < 0.05 and is deemed statistically significant. ** indicates *p* < 0.01 and is deemed statistically highly significant.

**Table 1 ijms-24-01245-t001:** Strains and plasmids used in this study.

Strains/Plasmids	Relevant Characteristic	Source or Reference
C83902	wild-type ETEC K88ac LT^+^,STa^+^, STb^+^	[34]
H10407	wild-type ETEC, CFA/I LT^+^,ST^+^	[35]
1836-2	wild-type ETEC K88ac EAST1^+^	[18]
EcNc	Wild-type Nissle 1917 deletion of plasmids pMUT1 and pMUT2	[36]
2015	1836-2 carrying pBR-p*eltAB*	This study
2016	1836-2 carrying pBR-pLTB_Q3K_	This study
2017	1836-2 carrying pBR-pLTB_G33S_	This study
2018	1836-2 carrying pBR-pLTB_T47A_	This study
2019	1836-2 carrying pBR-pLTA_R192G_	This study
2020	1836-2 carrying pBR-pLTA_L211A_	This study
2021	1836-2 carrying pBR-pLTA_R192G/L211A_	This study
2022	EcNc carrying pBR-h*eltAB*	This study
2023	EcNc carrying pBR-hLTB_Q3K_	This study
2024	EcNc carrying pBR-hLTB_G33S_	This study
2025	EcNc carrying pBR-hLTB_T47A_	This study
2026	EcNc carrying pBR-hLTA_R192G_	This study
2027	EcNc carrying pBR-hLTA_L211A_	This study
2028	EcNc carrying pBR-hLTA_R192G/L211A_	This study
Plasmid		
p2001	*estAB* from gene *E. coli* H10407 in pBR322	This study
p2002	*estAB* from gene *E. coli* C83902 in pBR322	This study
p2003	hLTB_Q3K_ in pBR322	This study
p2004	pLTB_Q3K_ in pBR322	This study
p2005	hLTB_G33S_ in pBR322	This study
p2006	pLTB_G33S_ in pBR322	This study
p2007	hLTB_T47A_ in pBR322	This study
p2008	pLTB_T47A_ in pBR322	This study
p2009	hLTA_R192G_ in pBR322	This study
p2010	pLTA_R192G_ in pBR322	This study
p2011	hLTA_L211A_ in pBR322	This study
p2012	pLTA_L211A_ in pBR322	This study
p2013	hLTA_R192G/L211A_ in pBR322	This study
p2014	pLTA_R192G/L211A_ in pBR322	This study

p means porcine, and h means human.

**Table 2 ijms-24-01245-t002:** The primer oligonucleotide sequences used in this study.

Name	Primer Sequences (5′-3′)
*ΔeltAB*-F	ATGAAAAATATAGCTTTCATTTTTTTTATTTTATTAGCATCGCCATTATATGCAAATTGTGTAGGCTGGAGCTGCTTCG
*ΔeltAB*-R	TGTTATATAGGTTCCTAGCATTAGACATGCTTTTAAAGCAAACTAGTTTTTCATACTCATATGAATATCCTCCTTAG
pBR-LT-F	CGGATTGTCTTCTTGTATGAT
pBR-LT-R	GATCGGTATTGCCTCCTCTAC
LTB_Q3K_-F	CACGGAGCTCCCAAAACTATTACAGAA
LTB_Q3K_-R	TTCTGTAATAGTTTTGGGAGCTCCGTG
LTB_G33S_-F	GAATCGATGGCATCCAAAAGAGAAATG
LTB_G33S_-R	CATTTCTCTTTTGGATGCCATCGATTC
LTB_T47A_-F	AAGAGCGGCGAAGCATTTCAGGTCGAA
LTB_T47A_-R	TTCGACCTGAAATGCTTCGCCGCTCTT
LTA_R192G_-F	GGAAATTCATCAGGAACAATTACAGG
LTA_R192G_-R	CCTGTAATTGTTCCTGATGAATTTCC
LTA_L211A_-F	AGCACAATATATGCCAGGGAATATCAA
LTA_L211A_-R	TTGATATTCCCTGGCATATATTGTGCT
FaeG-F	GGGAGCTGCTTTCGCTTTTT
FaeG-R	CCTCGGCAAACCACCATAAA
fimA-F	TGAATAACGGAACCAACACCATT
fimA-R	CGGCACCGGTTGCAA
FosA-F	CAGGCGGTTTACTACGCAACT
FosA-R	CGTCGGCGTTGGCAATA
fliC-F	CGCGGTCACCAACCTGAACAAC
fliC-R	CCTGCTGGATGATCTGCGCTT
GAPDH-F	CGTTAAAGGCGCTAACTTCG
GAPDH-R	ACGGTGGTCATCAGACCTTC
CysG-F	GGCAAGGGACGTTTGAAGAC
CysG-R	CGGCGATGACCAACCAA

## Data Availability

The data that support the findings of this study are available from the corresponding author upon reasonable request.

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
