# Peer review of "Both LTA and LTB Subunits Are Equally Important to Heat-Labile Enterotoxin (LT)-Enhanced Bacterial Adherence"

_ijms, 2023, doi:10.3390/ijms24021245_

Round 1
Reviewer 1 Report (Previous Reviewer 2)
I like the addition of the protein levels in of LT and its mutants in Figure 1. However, it is not clearly stated how the authors detected/ calculated the data presented in Figure 1F. Please add a corresponding description in the Method section and clearly state how many replicates were used to generate the data.
In addition, all figures still need further improvement. Most of them still appear pixilated and some panel indicators are cut off or are missing completely (E.g. Figure 4 Line 204, Figure 6 Line 264, Figure 7 Line 286.
Author Response
1.I like the addition of the protein levels in of LT and its mutants in Figure 1. However, it is not clearly stated how the authors detected/ calculated the data presented in Figure 1F. Please add a corresponding description in the Method section and clearly state how many replicates were used to generate the data.
Response: The ImageJ version 1.47v software (Wayne Rasband, National Institutes of Health, MD, USA) was used to quantify the images. The relative protein levels of LT and its mutants were calculated. The protein level of LT of 2015 strain was assumed as 100%, and all LT mutants were normalized to LT of 2015 strain. Data are expressed as the mean ± SD of three independent experiments. The information was added in the Method section in the revised manuscript.
2.In addition, all figures still need further improvement. Most of them still appear pixilated and some panel indicators are cut off or are missing completely (E.g. Figure 4 Line 204, Figure 6 Line 264, Figure 7 Line 286.
Response: All the figures have improved their qualities and re-uploaded in the submission system.
Reviewer 2 Report (Previous Reviewer 1)
Good
Author Response
No comment needs to response.
Reviewer 3 Report (New Reviewer)
1. Please write full forms of all the abbreviations at first time when used in manuscript
2.E.Coli italicized
3. If possible make a flow diagram of the methodology followed stepwise for increasing understanding and readibility for readers
Author Response
1. Please write full forms of all the abbreviations at first time when used in manuscript.
Response: All the abbreviations are defined the first time they appear in the revised manuscript.
2. E. Coliitalicized
Response: The “E. coli” is written in italics in the full revised manuscript.
3. If possible make a flow diagram of the methodology followed stepwise for increasing understanding and readibility for readers
Response: Thank you very much for your valuable suggestion. All experimental methods have been described in detail in the manuscript. I think the readers can understand them. Moreover, it isn’t easy to make the flow diagram.
This manuscript is a resubmission of an earlier submission. The following is a list of the peer review reports and author responses from that submission.
Round 1
Reviewer 1 Report
Overall this is an important investigation in the field. The experiments were set up well and is a great first step to show the importance of each subunit in adherence. However, this does need a better proofread. I would also suggest trying to think of a better way to present the strains as it is all quite confusing and you important findings get lost in the “and what was this strain again”.
Line 17: Awkward phrasing “evidences”, a suggestion would be “There is increasing evidence indicating that…….within in vitro….”
Line 19: Change “to” to “in” “…subunits in LT-enhanced..”
Line 20: Suggestion a word change of “LT mutants that their” changes to “LT mutants where their”
Line 22: place a bracket around 1836-2 and EcNc
Line 23: is numbers correct? “greater numbers of adhesion” awkward.
Line 23-27: please rephase the whole sentence. Ie “Moreover, E coli … strains when exogenously supplied…”
Line 27-28: The cholesterol experiment is really important but is not well introduced here. I would suggest adding a sentence before this one on what and why you did the cholesterol test and then the results would be more impressive. Currently the statement feels out of place.
Line 28-31: The “what” you did is lacking in the abstract and makes a sentence like this feel like an overstatement. Please briefly explain the experiences you conducted before making the results claim. You did not directly test and prove that a “OMV-LTB-GM1” bridge was form. Either remove or further justify this statement.
Line 36-38: Re word entire sentence, its too long and jumpy. Maybe split into two.
Line 39 and throughout: Can I suggestion that you change adhesins to adhesion molecules? And you deal with a large number of molecules, cells and a combination of each, its really important to be really clear and I think that could help improve clarity.
Line 48: Change “pathogenesis has been well known” to “is well established”
Line 52: Terming the A1 domain as a fragment is not the best descriptive factor. Could globular structure/polypeptide/domain be used?
Line 53: Sating that the A2 polypeptide “links” the A1 and LTB domains is again not the best word choice. That could mean covalent linkage, which it is not. Please change the word or add further clarification. Furthermore, saying “links the A1 fragment and the LTB subunits to form a stable hexamer” implies that the A1 is responsible the LTB stability, which is untrue. Please re-word.
Line 61: Remove “the” from “from both the in vitro”. Can citations also be included into that sentence.
Line 63: Change “stains” to “strains”
Line 72: I’d suggest changing required to important.
Line 82 onwards: Results section seem to be lacking a space between the word figure and the number.
Line 86-88: When talking about the cAMP levels it will elude to the experiment a bit better if you include “following incubation with X cells”. Currently it just reads that you have cAMP and the LTA protein. Where is the cAMP coming from.
Line 88: Change “induced by the three mutants” to “induced following incubation with X cells and X mutants”. Again need to mention cells and not over claim anything. What you measured was increased cAMP following incubation with cells and protein. That’s what needs to be said in the results, direct facts.
Line 94: Change “that” to “it” in “’’while that reduced approximately…”
Line 96: add a space between strains and 1836-2? Is that sentence meant to say “EcNc expression either native LT” or “EcNc expressed neither native”? Currently it’s a bit awkward.
Line 100 onwards: All the figure legends have “(A),” where it needs to be “(A),”
Figure 1: Can DMEM be define in the legend or have DMEM medium in the axis label, that would make it a lot clearer on what sort of control it is.
Figure 1 C and D could be supplementary data? Also, the X-axis numbers don’t need to be on an angle.
Line 100-101: Include the cells incubated in the description of cAMP.
Line 101: Change bingding to binding.
Line 105: A and B have error bars and that needs to be said within this line. Do C and D need error bars? Are they just done once.
Line 109 and 114: Space between the 2 and Δ for C83902ΔeltAB
Line 116: Including brackets and strain numbers/ details when saying “complemented” and wt strains would be beneficial.
Line 118: Can intestinal epithelial cells be said earlier in the paragraph, perhaps in line 110?
Line 106-118: Can a description of C83902 eltAB/pLT be included in the text?
Line 120: The opening lines of all the figures have the wrong tone. Don’t make the statement sound like “X was found”, change it to a direct and factual statement of what was done. Ie Fig 2 needs to be “Adherence percentage of differing ETEC cells to host cells” – you can adjust to your liking. Please change all accordingly. Furthermore, all axis titles are vague. I’d suggest to go over and see if any can be change to provide more precise information.
Line 128: Space between E. coli and 1836-2
Line 134: Avoid the use of “about”. Change to approximately. Please adjust throughout.
Line 134 – 135: I would suggest including some further background info in these strains when their differences are important. You do it in line 140.
Line 155: Include the word inhibitor before “Rp-cAMP” to help the reader understand what’s happening.
Figure 4/ line 159: The description for A is not the right tone. Change from what the data is telling the reader to what the experiment was.
Subheadings 2.4 and 2.5 are too similar. Either change the heading or merge that paragraphs.
Line 167: Briefly restate what FaeG ect, are again. Make it nice and clear to the reader.
Line 169 and 177: Change “adhesins” to “adhesin molecules”. Adjust sentence structure if needed.
Line 176-178: This sentence is an overstatement. Can reword to say “the first study that suggests that cAMP is important/vital to bacterial adherence”. Ie “may be the main reason” should be removed or changed.
Line 180-181: Change the tone as I’ve stated above. Facts not claims.
Line 187: Can “importance of LTB subunit” Be changed to “importance of the LTB subunit”
Line 189 and 192: Remove the odd spacing/new paragraphs.
Line 194: The LPS needs to be mentioned before this line. You need to better introduce this experiment by leading the reader into what was done. What controls were present to be able to state that it was LPS mediated?
Line 205 or 207: Can flow cytometry be mentioned around here to indicate how cholesterol levels were obtained.
Line 212: Reduc needs to change to reduce.
Line 213: Overstatement. You tested a marker of lipid rafts not directly lipid rafts. Please change to a factual statement.
Figure 7: Define MFI. Again wrong tone for the legend.
Line 224: English expression is poor for this opening line. “Despite it being well established that LT plays a role in enhancing enteric bacterial adherence, the precise molecular mechanisms of its two subunits remains to be determined”
Line 226-229: Mention of previous study but there is no citation. Please add.
Line 233: Change from “the LT mutants had a significantly reduced in its ability” to “the LT mutants had a significantly reduced ability..”
Line 248: The citation number is on the wrong side of the full stop/period.
Line 250: change expressed to expressing.
Line 225: remove “the” from “required for LTA mediated the enhancement”
Line 255 – 257: incorrect English expression. I’d suggest changing “given” to “the”. Or re-word the sentence to your liking. Perhaps combine it will the following sentence.
Line 264: Change treated to treatment
Line 264 -266: Poor English expression. Please proof-read again.
Line 266: Can “the cAMP level and up-regulate bacterial adhesins” change to “cAMP levels, which leads to up-regulated expression of bacterial adhesion molecules”
Line 267: Change “pathogens attached” to “pathogen attachment”
Line 269: Is this supposed to say “typical internalization pathway”?
Line 273: Change pre-treated to pre-treatment
Line 275: For the readers understanding, this sentence could be moved to the introduction. It would allow the reader to better understand why you completed the LPS experiment.
Line 273 – 276: You can’t start a sentence with “while” and then go nowhere with it. I’d suggest merging the two sentences.
Line 277: Can the mutation sites be included in this sentence? Re-iterated that you did the 3 and 47 mutations along with the 33.
Line 284: You need to say “with known impairment” since you did not directly test LPS binding.
Line 286 – 2293: Poor English expression in most of these sentences. Proof read.
Line 292: Change MOV to OMV
Line 292: Can more evidence and citations be included for the bridge comment.
Line 294: Remove one of the “results”
Line 299 – 301: Poor English expression. Proof read.
Discussion: The findings related to the cholesterol sequestering needs to be added into the discussion. You need to discuss lack of difference adherences for the 2017/G33 strain.
Line 318: Change “grown routinely” and provide a citation or methods.
Line 319: there appears to be a double space between LT and derivatives
Line 320 and later: You later define ampicillin as amp. Please be consistent.
Line 323 and later: Please include (w/w), (w/v) or (v/v) for all percentages.
Table 1: This is not very easy to understand. The first column is titled plasmids and strains and then only has plasmids as a subheading. Considering you only mention the strains in the text can that be the only factor in the left hand column and then include another column that contains plasmid information? Just re-think the lay out. If the number code for the strains was developed in this study is there any way to rename than and make them more reader friendly? Ie the strain name alludes to the cell line and the mutation it contains. What does h and p mean?
Line 329: Change overlap to overlapping
Line 330: Remove the s from double mutants
Line 342: Change “strains grown” to “strains were grown”
Line 342: Include “using LB” at the end of the sentence.
Line 346: Change “Likely” to “Additionally”
Line 349: Include what the lysis buffer contained.
Line 360: Change “Following” to “subsequently”
Line 360 – 362: Did a washing step occur between this two incubations? Please include.
Line 362: Perhaps start the sentence with “Then” instead of the number.
Line 367: “Bacterial cells were grown” is very vague. Can you include that they contain transfected plasmids? Or what cell line you are talking about.
Line 375: Change “that homologous” to “that is homologous” and change “regions of elt…” to “regions of the elt..”
Line 379: Change “Successfully deletion of elt..” to “Successful delection of the elt..”
Line 395: The respectively is confusing here. Please re-word
Line 416: Is removed the best word for this? Perhaps sequested
Line 431: What is meant by single cell?
Line 441: Is the cholesterol purchased? Where from?
Author Response
Point-by-point response to the reviewer’s comments
Submission ID: IJMS-1942667
Dear reviewer,
We are truly grateful to your suggestions to strengthen our manuscript with the valuable comments and queries. We have done much work to improve the manuscript according to the comments and suggestions. All the changes in the revised manuscript have been highlighted in yellow. The following is a point-by-point response to the questions and comments.
Reviewer 1#
- Overall this is an important investigation in the field. The experiments were set up well and is a great first step to show the importance of each subunit in adherence. However, this does need a better proofread. I would also suggest trying to think of a better way to present the strains as it is all quite confusing and you important findings get lost in the “and what was this strain again”.
Response: Thank you for your valuable suggestion. The full manuscript has been proofread to improve the quality of the manuscript. In order to present the strains clearly, the details of the strains were added after the strain numbers in the text. In addition, the plasmid information was provided in the Table 1.
- Line 17: Awkward phrasing “evidences”, a suggestion would be “There is increasing evidence indicating that…….within in vitro….”
Response: The sentence has been revised according to your suggestion.
- Line 19: Change “to” to “in” “…subunits in LT-enhanced..”
Response: Done.
- Line 20: Suggestion a word change of “LT mutants that their” changes to “LT mutants where their”
Response: Done.
- Line 22: place a bracket around 1836-2 and EcNc
Response: Done.
- Line 23: is numbers correct? “greater numbers of adhesion” awkward.
Response: Done.
- Line 23-27: please rephase the whole sentence. Ie “Moreover, E coli … strains when exogenously supplied…”
Response: The sentence has been rephrased to “Moreover, E. coli 1836-2 and EcNc strains when exogenously supplied with cAMP highly up-regulated the adhesins expression and improved their adherence abilities.”
- Line 27-28: The cholesterol experiment is really important but is not well introduced here. I would suggest adding a sentence before this one on what and why you did the cholesterol test and then the results would be more impressive. Currently the statement feels out of place.
Response: Thank you for your nice suggestion. Lipid rafts are small and dynamic regions enriched in cholesterol, Sphingolipids and ganglioside GM1, which participate in LT function. Moreover, ganglioside GM1 is the most important receptor for the B subunit of LT enterotoxin. In order to examine whether lipid rafts are required for LTB subunit-mediated enhancement of bacterial adherence, we carried out cholesterol experiment. The sentence “Ganglioside GM1, the receptor for LTB subunit, is enriched in lipid rafts.” has been added to introduce the aim of the experiment.
- Line 28-31: The “what” you did is lacking in the abstract and makes a sentence like this feel like an overstatement. Please briefly explain the experiences you conducted before making the results claim. You did not directly test and prove that a “OMV-LTB-GM1” bridge was form. Either remove or further justify this statement.
Response: The statement has been rephrased to “while LTB subunit mainly by mediating the initial interaction with the GM1 receptors of host cells.”
- Line 36-38: Re word entire sentence, its too long and jumpy. Maybe split into two.
Response: According to your suggestion, the long sentence has been split into two sentences “Enterotoxigenic Escherichia coli (ETEC) …in developing countries” and “In addition… young farm animals” in the revised manuscript.
- Line 39 and throughout: Can I suggestion that you change adhesins to adhesion molecules? And you deal with a large number of molecules, cells and a combination of each, its really important to be really clear and I think that could help improve clarity.
Response: “adhesins” has been replaced with “adhesin molecules” in the revised manuscript.
- Line 48: Change “pathogenesis has been well known” to “is well established”
Response: Done.
- Line 52: Terming the A1 domain as a fragment is not the best descriptive factor. Could globular structure/polypeptide/domain be used?
Response: The “A1 fragment” has been changed to “A1 polypeptide” in the revised manuscript.
- Line 53: Sating that the A2 polypeptide “links” the A1 and LTB domains is again not the best word choice. That could mean covalent linkage, which it is not. Please change the word or add further clarification. Furthermore, saying “links the A1 fragment and the LTB subunits to form a stable hexamer” implies that the A1 is responsible the LTB stability, which is untrue. Please re-word.
Response:The sentence has been rephrased to “The A2 polypeptide embeds the A1 polypeptide into the center of the LTB pentamericoligomer. LTA subunit is responsible for stabilizing the LT holotoxin structure and accelerating LTB subunit pentamer formation.” in the revised manuscript.
- Line 61: Remove “the” from “from both the in vitro”. Can citations also be included into that sentence.
Response: The word “the” has been removed, and the references 13-16 have been added to the sentence.
- Line 63: Change “stains” to “strains”
Response: Done.
- Line 72: I’d suggest changing required to important.
Response: “required” has been changed to “important”.
- Line 82 onwards: Results section seem to be lacking a space between the word figure and the number.
Response: A space has been added between the word figure and the number in the full results section.
- Line 86-88: When talking about the cAMP levels it will elude to the experiment a bit better if you include “following incubation with X cells”. Currently it just reads that you have cAMP and the LTA protein. Where is the cAMP coming from.
Response: The sentence has been rephrased to “In contrast to the native LT, … mutants following incubation with IPEC-J2 cells were… (Figure 1A)” in the revised manuscript.
- Line 88: Change “induced by the three mutants” to “induced following incubation with X cells and X mutants”. Again need to mention cells and not over claim anything. What you measured was increased cAMP following incubation with cells and protein. That’s what needs to be said in the results, direct facts.
Response: Thank you for your valuable suggestion. According to your suggestion, the sentence has been changed to “The levels of cAMP induced following incubation with IPEC-J2 cells and LTAR192G, LTAL211A and dmLT mutants…” in the revised manuscript.
- Line 94: Change “that” to “it” in “’’while that reduced approximately…”
Response: Done.
- Line 96: add a space between strains and 1836-2? Is that sentence meant to say “EcNc expression either native LT” or “EcNc expressed neither native”? Currently it’s a bit awkward.
Response: A space has been added between strains and 1836-2. The sentence has been changed to “EcNc expression native LT or LT mutants….” In the revised manuscript.
- Line 100 onwards: All the figure legends have “(A),” where it needs to be “(A),”
Response: This errors have been changed in the full revised manuscript.
- Figure 1: Can DMEM be define in the legend or have DMEM medium in the axis label, that would make it a lot clearer on what sort of control it is.
Response: “The intracellular level of cAMP from IPEC-J2 cells just incubation with DMEM medium was used as the negative control.” has been addressed in the figure legend.
- Figure 1 C and D could be supplementary data? Also, the X-axis numbers don’t need to be on an angle.
Response: Thank you for your suggestion, but I think that figure 1C and 1D may be better to show in the manuscript text as the parts of figure 1. The angles of the X-axis numbers have been removed in the re-drafted figures.
- Line 100-101: Include the cells incubated in the description of cAMP.
Response: “following incubation with IPEC-J2 cells” has been added.
- Line 101: Change bingding to binding.
Response: Done.
- Line 105: A and B have error bars and that needs to be said within this line. Do C and D need error bars? Are they just done once.
Response: “Data are expressed as the mean ± standard deviation of triplicate experiments.” has been said within this line. The bacterial growth curves data are presented as the mean ± standard deviation of quadruplicate independent experiments, and the error bars are added in the re-drafted figure 1C and 1D.
- Line 109 and 114: Space between the 2 and ΔforC83902ΔeltAB
Response: Space between the 2 and Δ for C83902ΔeltAB and space between the 7 and Δ for H10407ΔeltAB were added among 109-114 lines.
- Line 116: Including brackets and strain numbers/ details when saying “complemented” and wt strains would be beneficial.
Response: The details have been added in the revised manuscript.
- Line 118: Can intestinal epithelial cells be said earlier in the paragraph, perhaps in line 110?
Response: Done.
- Line 106-118: Can a description of C83902 eltAB/pLT be included in the text?
Response: C83902 ΔeltAB/pLT complemented strain is C83902 ΔeltAB isogenic mutant containing a recombinant plasmid expressing the full eltAB gene. The information has been added.
- Line 120: The opening lines of all the figures have the wrong tone. Don’t make the statement sound like “X was found”, change it to a direct and factual statement of what was done. Ie Fig 2 needs to be “Adherence percentage of differing ETEC cells to host cells” – you can adjust to your liking. Please change all accordingly. Furthermore, all axis titles are vague. I’d suggest to go over and see if any can be change to provide more precise information.
Response: Thank you for your nice suggestion. The opening lines of all the figures have been changed to a direct and factual statement. All the axis titles are also changed to provide more precise information in the re-drafted figures.
- Line 128: Space between E. coli and 1836-2
Response: A space has been added.
- Line 134: Avoid the use of “about”. Change to approximately. Please adjust throughout.
Response: “about” have been changed to “approximately” through the full revised manuscript.
- Line 134 – 135: I would suggest including some further background info in these strains when their differences are important. You do it in line 140.
Response: The background information of these strains has been added in the revised manuscript.
- Line 155: Include the word inhibitor before “Rp-cAMP” to help the reader understand what’s happening.
Response: Done.
- Figure 4/ line 159: The description for A is not the right tone. Change from what the data is telling the reader to what the experiment was.
Response: The description for A and B has been changed to clarify the experiment.
- Subheadings 2.4 and 2.5 are too similar. Either change the heading or merge that paragraphs.
Response: The subheading 2.5 has been changed to “cAMP increases the expression of bacterial adhesin molecules” in the revised manuscript.
- Line 167: Briefly restate what FaeG ect, are again. Make it nice and clear to the reader.
Response: faeG (encoding the major fimbrial subunit of F4 fimbriae), fliCH4 (encoding the flagellin H4 serotype), fimA (encoding the major fimbrial subunit of type I fimbriae), focA (encoding the major fimbrial subunit of F1C fimbriae), fliCH1 (encoding the flagellin H1 serotype), the information was added in the revised manuscript.
- Line 169 and 177: Change “adhesins” to “adhesin molecules”. Adjust sentence structure if needed.
Response: Done.
- Line 176-178: This sentence is an overstatement. Can reword to say “the first study that suggests that cAMP is important/vital to bacterial adherence”. Ie “may be the main reason” should be removed or changed.
Response: This sentence has been re-phrased to “These results suggested that cAMP is important to bacterial adherence.”
- Line 180-181: Change the tone as I’ve stated above. Facts not claims.
Respons: The opening line of all the figure.5 has been changed to “Exogenously supplied 5 μM and 10 μM cAMP up-regulates the adhesin molecules mRNA expression of E. coli 1836-2 and EcNc.”
- Line 187: Can “importance of LTB subunit” Be changed to “importance of the LTB subunit”
Response: The word “the” was added.
- Line 189 and 192: Remove the odd spacing/new paragraphs.
Response: The spacing has been removed.
- Line 194: The LPS needs to be mentioned before this line. You need to better introduce this experiment by leading the reader into what was done. What controls were present to be able to state that it was LPS mediated?
Response: “Previous studies have demonstrated that the Gly-33 residue of LTB subunit is required for binding to GM1 receptors, while the Gln-3 and Thr-47 residues are required for binding to LPS. To explore the importance of the GM1 and LPS binding abilities of LTB subunit for bacterial adherence,…” the introduction and aim of this experiment are added at the beginning of the paragraph.
It has been demonstrated that the Gln-3 and Thr-47 residues are important for LTB and LPS interaction, compared to the native LT enterotoxin, the ability to enhance E. coli 1836-2 and EcNc strains significantly decreased. The results indicated that the LPS binding ability is important for LTB subunit mediated enhancement of bacterial adherence. The controls are 1836-2/pLT and EcN/hLT strains, respectively.
- Line 205 or 207: Can flow cytometry be mentioned around here to indicate how cholesterol levels were obtained.
Response: This has been mentioned in the revised manuscript.
- Line 212: Reduc needs to change to reduce.
Response: “Reduc” has been changed to “reduce”
- Line 213: Overstatement. You tested a marker of lipid rafts not directly lipid rafts. Please change to a factual statement.
Response: The statement has been rephrased to “These results suggested that the GM1 binding ability and membrane cholesterol play a crucial role in LTB-enhanced bacterial adherence.”
- Figure 7: Define MFI. Again wrong tone for the legend.
Response: MFI is mean fluorescence intensity and has defined in the figure.7 legend. The opening line of all the figure.7 has changed to “Membrane cholesterol is important for LT-increased ETEC 1836-2 adhesion to IPEC-J2 cells.”
- Line 224: English expression is poor for this opening line. “Despite it being well established that LT plays a role in enhancing enteric bacterial adherence, the precise molecular mechanisms of its two subunits remains to be determined”
Response: Thank you, the sentence has been rephrased according to your suggestion.
- Line 226-229: Mention of previous study but there is no citation. Please add.
Response: The references 13-15 have been cited.
- Line 233: Change from “the LT mutants had a significantly reduced in its ability” to “the LT mutants had a significantly reduced ability.”
Response: Done.
- Line 248: The citation number is on the wrong side of the full stop/period.
Response: Done
- Line 250: change expressed to expressing.
Response: “expressed” has been changed to “expressing”.
- Line 255: remove “the” from “required for LTA mediated the enhancement”
Response: The word “the” has been removed.
- Line 255 – 257: incorrect English expression. I’d suggest changing “given” to “the”. Or re-word the sentence to your liking. Perhaps combine it will the following sentence.
Response: “Given” has been changed to “The” in the revised manuscript.
- Line 264: Change treated to treatment
Response:“treated” has been changed to “treatment”.
- Line 264 -266: Poor English expression. Please proof-read again.
Response: The sentence has been rephrased.
- Line 266: Can “the cAMP level and up-regulate bacterial adhesins” change to “cAMP levels, which leads to up-regulated expression of bacterial adhesion molecules”
Response: Thank you, the sentence has been changed to “cAMP levels, which leads to up-regulated expression of bacterial adhesin molecules”.
- Line 267: Change “pathogens attached” to “pathogen attachment”
Response: Done.
- Line 269: Is this supposed to say “typical internalization pathway”?
Response: Yes. We have changed in the revised manuscript.
- Line 273: Change pre-treated to pre-treatment
Response: “pre-treated” has been changed to “pre-treatment”.
- Line 275: For the readers understanding, this sentence could be moved to the introduction. It would allow the reader to better understand why you completed the LPS experiment.
Response: Thank you for your nice suggestion. The content “A previous …while the Gln-3 and Thr-47 residues are required for binding to LPS (13-16).” has been moved to the introduction part.
- Line 273 – 276: You can’t start a sentence with “while” and then go nowhere with it. I’d suggest merging the two sentences.
Response: The two sentences have been merged to one sentence and moved to the introduction part.
- Line 277: Can the mutation sites be included in this sentence? Re-iterated that you did the 3 and 47 mutations along with the 33.
Response: The mutation sites of LTB subunit have been mentioned in this sentence.
- Line 284: You need to say “with known impairment” since you did not directly test LPS binding.
Response: “with strong impairment” has changed to “with known impairment”.
- Line 286 – 293: Poor English expression in most of these sentences. Proof read.
Response: Thank you for your suggestion. The content has been rephrased to “Therefore, the LTB subunit contributes to increasing bacterial adherence, which predominantly depends on its GM1 receptor binding activity, but also on its LPS binding activity. LTB subunit could regulate LTB-enhanced bacterial adherence in two different ways. First, the interaction of LTB subunit with GM1 receptors on the surface of host cells facilitates the internalization of LTA subunit in which LTB exerts its enterotoxicity. Besides, the LTB subunit binding to the LPS in OMVs located on the bacterial surface would form an “OMVs-LTB-GM1” bridge between the bacteria and host cells to improve subsequent adherence.” in the revised manuscript.
- Line 292: Change MOV to OMV
Response: Done.
- Line 292: Can more evidence and citations be included for the bridge comment.
Response: The references 11 and 19 were cited here.
- Line 294: Remove one of the “results”
Response: One of the word “results” has been removed.
- Line 299 – 301: Poor English expression. Proof read.
Response: The sentence has been rephrased to “The released cAMP elicited by LT can regulate the expression of bacterial adhesion molecules, in turn suggesting a cross-talk in the expression of virulence determinants when pathogens cause infections.” in the revised manuscript.
- Discussion: The findings related to the cholesterol sequestering needs to be added into the discussion. You need to discuss lack of difference adherences for the 2017/G33 strain.
Response: The content has been mentioned in the discussion part in the revised manuscript.
- Line 318: Change “grown routinely” and provide a citation or methods.
Response: “grown routinely” has been changed to “cultured”, and the references 18, 34 and 36 were cited.
- Line 319: there appears to be a double space between LT and derivatives
Response: The excess space has been removed.
- Line 320 and later: You later define ampicillin as amp. Please be consistent.
Response: Amp was used in the later.
- Line 323 and later: Please include (w/w), (w/v) or (v/v) for all percentages.
Response: The 10% heat-inactivated fetal bovine serum is v/v, 5% CO2 is v/v, and 0.25% trypsin is m/v. All these have been mentioned in the revised manuscript.
- Table 1: This is not very easy to understand. The first column is titled plasmids and strains and then only has plasmids as a subheading. Considering you only mention the strains in the text can that be the only factor in the left hand column and then include another column that contains plasmid information? Just re-think the lay out. If the number code for the strains was developed in this study is there any way to rename than and make them more reader friendly? Ie the strain name alludes to the cell line and the mutation it contains. What does h and p mean?
Response: The plasmid information was added in the Table 1. To make them easy to understand, the details of the strains were added after the strain numbers in the text. “h” means human, and “p” means porcine. These have been mentioned under Table 1.
- Line 329: Change overlap to overlapping
Response: Done.
- Line 330: Remove the s from double mutants
Response: “s” has been removed.
- Line 342: Change “strains grown” to “strains were grown”
Response: “strains grown” has been changed to “strains were grown”.
- Line 342: Include “using LB” at the end of the sentence.
Response: This has been added at the end of the sentence.
- Line 346: Change “Likely” to “Additionally”
Response: Done.
- Line 349: Include what the lysis buffer contained.
Response: The cell lysis buffer (PART#895890) is provided in the cAMP Parameter assay kit (R&D Systems, USA), however, more detailed information on this lysis buffer is notprovided in the kit.
- Line 360: Change “Following” to “subsequently”
Response: “Following” has been changed to “Subsequently”.
- Line 360 – 362: Did a washing step occur between this two incubations? Please include.
Response: Yes. The plates were washed three times with PBS+0.05% (v/v) Tween 20 (PBST), and the content was added in the revised manuscript.
- Line 362: Perhaps start the sentence with “Then” instead of the number.
Response: The word “Then” was added at the beginning of the sentence.
- Line 367: “Bacterial cells were grown” is very vague. Can you include that they contain transfected plasmids? Or what cell line you are talking about.
Response: The details of the bacteria (containing the transfected plasmids) were provided in the revised manuscript.
- Line 375: Change “that homologous” to “that is homologous” and change “regions of elt…” to “regions of the elt..”
Response: Done.
- Line 379: Change “Successfully deletion of elt..” to “Successful deletion of the elt..”
Response: The “Successfully deletion of elt..” has been changed to “Successful deletion of the elt..”.
- Line 395: The respectively is confusing here. Please re-word
Response: The sentence has been rephrased to “…each of these two strains was grown under LB medium containing 0, 2, 5, or 10 μM cAMP for 3 h to logarithmic growth phase.”
- Line 416: Is removed the best word for this? Perhaps sequested
Response: The word “the” was added.
- Line 431: What is meant by single cell?
Response: Single cell means individual cells, and there are no double cells or clustered cells. In order to be understood, “single cells” has been changed to “at 37°C for 20 min”
- Line 441: Is the cholesterol purchased? Where from?
Response: Yes, cholesterol is purchased from Sigma-Aldrich (CAS NO. 8667). The information was added in the revised manuscript.
Reviewer 2 Report
The manuscript „Both LTA and LTB subunits are equally important to heat-labile enterotoxin-mediated enhancement of mediated adherence” is a collection of different experiment to show how both LT subunits contribute to the adherence of the tested ETEC strains. All in all, I think that the study shows some potential, but the manuscript has several aspects that can be majorly improved before publications.
Some detailed comments:
1. Title: I think the term enhancement is wrong in this context
2. Introduction:
a. The usage of literature references is rather sparse and can be improved.
b. LT contains only on disulfide bond as all other AB5 toxins (LINE 52)
c. Motivation of the study is well described but not why performed with a porcine and human strain?
3. Results:
a. Different methods used to obtain different results are not indicated
b. No explanation given why exactly those mutants were chosen to study different effects
c. Figure 1: legend for columns in A and B missing; clarification on the strain numbers with respect to the mutations and strains used would be advantageous (reference to table 1?); C and D: y-axes log-values; labeling of x-axes why rotated?
d. Include the full strain name for the first time a strain mentioned (LINE 109, 111)
e. Missing clarification on porcine and human experimental aspect
f. Figure 3: how is the fold change calculated?
g. 2.4: What is the cAMP effect on mutants?
h. 2.5: state why those genes were chosen for qRT-PCR?; include negative control that is not effected by cAMP levels; include controls on housekeeping genes used for data evaluation (supplements)
i. Figure 5: Percentage of relative gene expression of what?
j. 2.7: motivation for lipid rafts experiment?
k. Figure 7: Missing legend. What is represented by the different fillings of the columns?
4. Discussion:
a. What is the effect of removing one subunit of LT? I am not convinced that both subunits play an equal role in the LT-mediated bacterial adherence. Removing one subunit at a time and compare the effect of this could answer the question of importance to the adherence of ETEC strains
b. I think the term enhanced with respect to the adherence of ETEC strains is not the right term.
c. Clarify the novelty of the presented results and this study. There are already studies that indicated that LTA and LTB contribute to the adherence. Also the importance of the disulfide bound on the A-subunit is well known. Clearer statements on the study concept and more detailed discussion of the obtained results is needed.
d. Why use a porcine and human strain, what differences are expected or not?
5. Methods:
a. 4.7. number of technical replicates?
b. 4.10. results obtained with the described method?
Author Response
Point-by-point response to the reviewer’s comments
Submission ID: IJMS-1942667
Dear reviewer,
We are truly grateful to your suggestions to strengthen our manuscript with the valuable comments and queries. We have done much work to improve the manuscript according to the comments and suggestions. All the changes in the revised manuscript have been highlighted in yellow. The following is a point-by-point response to the questions and comments.
Reviewer 2#
The manuscript, Both LTA and LTB subunits are equally important to heat-labile enterotoxin-mediated enhancement of mediated adherence” is a collection of different experiment to show how both LT subunits contribute to the adherence of the tested ETEC strains. All in all, I think that the study shows some potential, but the manuscript has several aspects that can be majorly improved before publications.
Response: We are truly grateful for your valuable comments and suggestions. We have done much work to improve the manuscript according to the comments and suggestions. All the changes in the revised manuscript have been highlighted in yellow. The following is a point-by-point response to your questions and comments.
- Title: I think the term enhancement is wrong in this context
Response: The title has been changed to “Both LTA and LTB subunits are equally important to heat-labile enterotoxin (LT)-enhanced bacterial adherence” in the revised manuscript.
- The usage of literature references is rather sparse and can be improved.
Response: The full manuscript has been proofread, and more literature references were cited in which they are needed. The new cited references in the text have been highlighted in yellow in the revised manuscript.
- LT contains only on disulfide bond as all other AB5 toxins (LINE 52)
Response: Thank you for your information. The error has been corrected.
- Motivation of the study is well described but not why performed with a porcine and human strain?
Response: LT can be categorized into LTh derived from humans and LTp derived from piglets. LT gene was detected in 57.7% of ETEC isolates associated with porcine diarrhea in the US (Zhang et al., 2007). In addition, it was reported that approximately 60% of field ETEC isolates associated with human diarrhea expressed either LT alone (27%) or LT with ST (33%) (Isidean et al., 2011). The results suggested LT is a crucial virulence factorfor ETEC isolates derived from both porcine and human to induce diarrhea. To elucidate the precise mechanism of LTA and LTB subunits in promoting bacterial adherence we introduced plasmids that contains porcine or human LT and their mutants to a porcine (1836-2) and human (EcNc) strain, respectively. Our results indicated the mechanism of LTh and LTp to enhance bacterial adherence is identical. More importantly, the results from these two different strains can support each other.
- Different methods used to obtain different results are not indicated.
Response: Wang and his colleagues found that inhibiting the cAMP-dependent signal pathway blocked LT-increased bacterial adherence, suggesting that LT contributing to the improvement in adherence ability of ETEC is mainly dependent upon its LTA subunit (Wang et al., 2012). In contrast, another study found that the F4+ ETEC strain expressing the LTB subunit or the non-ADP-ribosylation activity mutant had a similar ability adherence to IPEC-J2 cells as the strain expressing LT holotoxin. Moreover, pre-treatment of IPEC-J2 cells cycloheximide had no influence on LT-enhanced bacterial adherence. The results suggested that the LTB subunit plays a predominant role in LT-promoted bacterial adherence (Fekete et al., 2013). The content has been addressed in the revised manuscript.
- No explanation given why exactly those mutants were chosen to study different effects.
Response: Some specific amino acid residues are necessary for the efficient function of LT. It has been reported that some monomeric mutant of the LTA subunit (S63K, A72R, R192G, L211A) highly impairs its enterotoxicity. In addition, another double mutant LTA subunit (R192G/L211A) was shown to just retain less than 0.1% of the enzymatic activity of LT. Previous studies suggested that the ADP-ribosylation activity of the LTA subunit is necessary for LT-enhanced bacterial adherence. To better clarify the aim of our study and why we chose these LT mutants, the content has been removed from the discussion part to the introduction part in the revised manuscript.
- Figure 1: legend for columns in A and B missing; clarification on the strain numbers with respect to the mutations and strains used would be advantageous (reference to table 1?); C and D: y-axes log-values; labeling of x-axes why rotated?
Response: The legend was added, and the Fig.1A and 1B were re-drafted according to your suggestion. In fact, the intracellular levels of cAMP induced by 2015 (1836-2/pLT), 2019 (1836-2/pLTAR192G), 2020 (1836-2/pLTAL211A), and 2021(1836-2/ pLTAR192G/L211A) strains following incubation with IPEC-J2 cells, and the intracellular levels of cAMP induced by 2022 (EcN/hLT), 2026 (EcN/hLTAR192G), 2027 (EcN/hLTAL211A) and 2028 (EcN/ hLTAR192G/L211A) following incubation with IPEC-J2 cells were measured. We just showed the data from IPEC-J2 cells because the data from Caco-2 cells is similar. We just showed the GM1-binding ELISA data from 2015, 2016 (1836-2/pLTBQ3K), 2017(1836-2/pLTBG33S) and 2018 (1836-2/pLTBT47A) strains as the same reason. The Fig.1C and 1D were re-drafted according to the suggestions too.
- Include the full strain name for the first time a strain mentioned (LINE 109, 111)
Response: The full name and detailed information of the strains were provided in the revised manuscript.
- Missing clarification on porcine and human experimental aspect
Response: No porcine and human experiments in our study.
- Figure 3: how is the fold change calculated?
Response: In the Fig.3A, the colony forming units (CFUs) of wild-type ETEC 1836-2 adhesion to IPEC-J2 cells was assumed as 100%. While in In the Fig.3B, the colony forming units (CFUs) of parent E. coli EcNc strain adhesion to Caco-2 cells was assumed as 100%. The content has mentioned in legend for figure.3.
- 2.4: What is the cAMP effect on mutants?
Response: We didn’t test the cAMP effect on the adherence ability of 1836-2 and EcNc strains expressing the LT mutants.
Our above results found that the negative-LT strains expressing the LTA mutants that impaired ADP enzymatic activity had a significant reduction in the ability to enhance bacterial adherence compared to the strains expressing the native LT, suggesting that the ADP enzymatic activity of LTA subunit is important for LTA-enhanced bacterial adherence. To confirm the results, we tested the adherence ability of two negative-LT strains (1836-2 and EcNc) after pre-incubating with exogenous cAMP. Our data showed that exogenously supplied cAMP leads to a significant increase in E. coli 1836-2 and EcNc strains adhesion to IPEC-J2 or Caco-2 cells in a concentration-dependent manner. I think our results are enough to support that the LTA subunit contributing to LT-enhanced bacterial adherence was dependent on its ability to elevate the level of cAMP.
- 2.5: state why those genes were chosen for qRT-PCR?; include negative control that is not effected by cAMP levels; include controls on housekeeping genes used for data evaluation (supplements)
Response: Bacteria express adhesion molecules to promote the initial interaction with host cell receptors, which is beneficial for subsequent adhesion and colonization. Thus, we speculated that the role of LTA in enhancing bacterial adherence may be mainly due to the increased expression of bacterial adhesion molecules. We tested the expression level of the main adhesion molecules genes of 1836-2 (faeG, fimA and fliCH4), and EcNc (fimA, focA and fliCH1) using RT-PCR method. The content has stated in result 2.5.
The glyceraldehyde-3-phosphate dehydrogenase (GAPDH) and cysG genes that were not affected by cAMP levels were chosen as the reference genes for data evaluation. The content has mentioned in method 4.8.
- Figure 5: Percentage of relative gene expression of what?
Response: Percentage of relative adhesin genes expression vs the expression level of these genes under 0μM cAMP treatment. That has stated in the new figure.5.
- 2.7: motivation for lipid rafts experiment?
Response: Lipid rafts are small and dynamic regions enriched in cholesterol, Sphingolipids and ganglioside GM1, which participate in LT function. Moreover, ganglioside GM1 is the most important receptor for the B subunit of LT enterotoxin. To examine whether lipid rafts are required for LTB subunit-mediated enhancement of bacterial adherence, we did the cholesterol experiment. The sentence “Ganglioside GM1, the receptor for LTB subunit, is enriched in lipid rafts.” has been added to introduce the aim of the experiment.
- Figure 7: Missing legend. What is represented by the different fillings of the columns?
Response: The different fillings of the columns of Fig.7A represent the cell viability of IPEC-J2 cells after treatment with 0, 1, 2, 4, 8, 10 and 15 mM MβCD, which has been included in the new figure.
- What is the effect of removing one subunit of LT? I am not convinced that both subunits play an equal role in the LT-mediated bacterial adherence. Removing one subunit at a time and compare the effect of this could answer the question of importance to the adherence of ETEC strains
Response: The functional holotoxin LT is comprised of a single LTA subunit and a pentameric LTB subunit. The complete structure of LT holotoxin is necessary for its efficient function. Previous studies investigated the role of LTA or LTB subunit in increasing bacterial adherence by deletion of one subunit at a time or expressing one subunit alone. They can’t reflect the function of the subunit in a physiological state due to this method can’t keep LT holotoxin in the form of A1B5. Some specific amino acid residues are necessary for the efficient function of LT. In our study, several LT mutants that impaired their ADP-ribosylation activity or receptor binding ability were constructed. Furthermore, the abilities of the native LT and LT derivatives to enhance bacterial adherence were evaluated in vitro cell models. Our data showed that the attenuated enterotoxicity LTA mutants and the impaired GM1 binding ability LTB mutants showed similar abilities to improve bacterial adherence, indicating both subunits play an equal role in LT-enhanced bacterial adherence.
- I think the term enhanced with respect to the adherence of ETEC strains is not the right term.
Response: Thank you for your nice suggestion. In fact, many studies have used the word “enhanced”concerning the adherence of ETEC strains (Johnson, et al., 2009; Wang, et al., 2012; Fekete et al., 2013). Therefore, we used the words “enhance” “increase” as well as “promote” in this study.
- Clarify the novelty of the presented results and this study. There are already studies that indicated that LTA and LTB contribute to the adherence. Also the importance of the disulfide bound on the A-subunit is well known. Clearer statements on the study concept and more detailed discussion of the obtained results is needed.
Response: Previous studies investigated the role of LTA or LTB subunit in promoting bacterial adherence mainly by deletion of one subunit at a time or expressing one subunit alone in the non-LT-producing strains. The results can’t exactly reflect the function of each subunit in a normal physiological state due to the inability to form the functional LT holotoxin. In this study, the single or double LT mutants were generated by site-directed mutagenesis of the key specific amino acid residues, which just impaired the efficient function of the LTA or LTB subunit, but would not affect the biological structure of the LT holotoxin. Therefore, the obtained results in our study may be more reliable than the previous reports. The statements have been included in the discussion part of the revised manuscript.
- Why use a porcine and human strain, what differences are expected or not?
Response: LT can be categorized into LTh derived from humans and LTp derived from piglets. LT gene was detected in 57.7% of ETEC isolates associated with porcine diarrhea in the US (Zhang et al., 2007). In addition, it was reported that approximately 60% of field ETEC isolates associated with human diarrhea expressed either LT alone (27%) or LT with ST (33%) (Isidean et al., 2011). The results suggested LT is a crucial virulence factor for ETEC isolates derived from both porcine and human to induce diarrhea. To elucidate the precise mechanism of LTA and LTB subunits in promoting bacterial adherence, we introduced plasmids that contain porcine or human LT and their mutants to a porcine (1836-2) and human (EcNc) strain, respectively. Our results indicated the mechanism of LTh and LTp to enhance bacterial adherence is identical. More importantly, the results from these two different strains can support each other.
- 4.7. number of technical replicates?
Response: Three technical replicates were used in each experiment, with three biological replicates.
- 4.10. results obtained with the described method?
Response: Flow cytometry analysis results showed that approximately 30% of cholesterol was depleted in IPEC-J2 cells after being treated with 10 mM MβCD. The results have been addressed in result 2.7 and shown in Fig.7B.
Reviewer 3 Report
The paper discusses how LTA and LTB subunits of heat-labile enterotoxin (LT) enhance the bacterial adherence. There are some important questions noticed during the review process.
1. The paper used two strains with no expression of LTA and LTB as models. Then reintroduce the protein expression back by carrying different plasmids. However, there is no estimation to show the yields of protein expression in the strains. If there is no (or less or various) expression, does the effect change the conclusion? Without the data, the result will not be convincing. More important, for the introducing LTA and/or LTB by plasmid, if comparing to endogenous LTA/LTB, the expression level would be comparable? or much higher than the LTA/LTB expression in normal enterotoxigenic E. coli (ETEC).
2. When the paper tried to connect the ADP-ribosylation activity and cAMP concentration, there lacks a connection how ADP-ribosylation in bacteria can enhance cellular cAMP concentration. In my understanding, the connection ADP-ribosylation activity and cAMP is based on the effect in “host cells”, not bacteria. In host cells, LTA modifies adenylyl cyclase. How about the corresponding mechanism in bacteria? Why we also need think ADP-ribosylation activity and cAMP in bacteria. Expect to have more explanation.
3. When the paper defined that LTA ADP-ribosylation activity will be critical for LT-derived adhesion and ADP-ribosylation activity is related to cAMP level. Strains of 2019, 2020, 2021 still caused certain increases in the bacterial adherence (Fig.3). However, based on the carried plasmid, the expressed LTA mutants might have less ADP-ribosylation activity and no enhancement of cAMP concentration (Fig. 1). Also need an explanation why mutated LTA also enhanced adherence.
4. In addition, the paper tried to connect cAMP to adhesins expression. However, besides cAMP-leading adhesins expression (Fig. 5), do the authors check the enhancement of adhesins expression in the LTA-expressed strains (even in LTB-expressed strains)? LTB affects cAMP and adhesins? If not, any possible mechanism to explain why LTB is critical for adherence.
5. Why there are no strains only carrying LTA or LTB. There are only strains carrying mutated LTA or LTB?
There are many other minor errors. Authors might consider to reorganize the paper. For example, it is difficult to understand so many strains in the first time of reading.
Round 2
Reviewer 2 Report
Dear Authors,
thank you for your detailed response letter. All conducted changes on the manuscript improved it and I really enjoyed reading it in this form.
The last thing I would suggest is to improve the resolution of the figures, as they appeared very pixilated and remove the beige background.
All the best.
Author Response
Reviewer 2#
1.The last thing I would suggest is to improve the resolution of the figures, as they appeared very pixilated and remove the beige background.
Response: Thank you very much for your valuable suggestion. All the figures have been re-drafted to remove the beige background and improve the resolution in the revised manuscript.

Reviewer 3 Report
With the improvement of the new version of manuscript, I will hope to see the information of protein expression level in the E. coli. There are many mutated proteins showing very different expression levels in the E. coil system.
Author Response
Reviewer3#
1.With the improvement of the new version of manuscript, I will hope to see the information of protein expression level in the E. coli. There are many mutated proteins showing very different expression levels in the E. coil system.
Response: Thank you very much for your nice suggestion. We don’t think it is necessary to directly detect the protein expression level of LT and LT mutants in E. coli 1836-2 and EcNc. The reasons for this are as follows. First, the production of LT enterotoxins mainly regulated by the strains’ growth conditions (PH, temperature, oxygen level, osmolarity). In this study, we cultured all the strains in the same growth condition, the results showed that the E. coli strains 1836-2 and EcNc expressing native LT or LT mutants had similar growth rates. This can rule out the influence of culture conditions on the expression of LT. Second, the native LT and LT mutants were expressed in the same host strains (1836-2 or EcNc). This can rule out the influence of different host strains on the production of LT. Third, the ADP-ribosylation of the LTA subunit and the GM1-binding abilities of the LTB subunits can’t reflect from the protein expression level in the two strains. In order to evaluate the role of LTA or LTB subunit in LT-enhanced bacterial adherence, we generated the LTAR192G, LTAL211A and dmLT mutants of LTA subunit, the LTBQ3K, LTBG33S and LTBT47A mutants of LTB subunit. Further, their ADP-ribosylation and the GM1-binding abilities were measured by cAMP ELISA and GM1-ELISA methods.
Fourth, we have used the 1836-2/pLT and EcNc/pLT strains as the positive controls, and the parent strains as the negative controls when carried out the cAMP ELISA and GM1-ELISA assays. In summary, our results indicated the decreased ability of LT mutants to enhance bacterial adherence was attributed to the impairment of the ADP-ribosylation and the GM1-binding activities, rather than due to the different protein expression levels.
